# Attentional priorities drive effects of time pressure on altruistic choice

Yi Yang Teoh [1✉], Ziqing Yao[2,3], William A. Cunningham[1,4] & Cendri A. Hutcherson [1,4]

Dual-process models of altruistic choice assume that automatic responses give way to deliberation over time, and are a popular way to conceptualize how people make generous choices and why those choices might change under time pressure. However, these models have led to conflicting interpretations of behaviour and underlying psychological dynamics. Here, we propose that flexible, goal-directed deployment of attention towards information priorities provides a more parsimonious account of altruistic choice dynamics. We demonstrate that time pressure tends to produce early gaze-biases towards a person's own outcomes, and that individual differences in this bias explain how individuals' generosity changes under time pressure. Our gaze-informed drift-diffusion model incorporating moment-to-moment eye-gaze further reveals that underlying social preferences both drive attention, and interact with it to shape generosity under time pressure. These findings help explain existing inconsistencies in the field by emphasizing the role of dynamic attention-allocation during altruistic choice.

[1] Department of Psychology, University of Toronto, Toronto, Canada. [2] Department of Psychology, The University of Hong Kong, Hong Kong, China. [3] School of Psychology, Center for Studies of Psychological Application and Key Laboratory of Mental Health and Cognitive Science of Guangdong Province, South China Normal University, Guangzhou, China. [4] Department of Marketing, Rotman School of Management, University of Toronto, Toronto, Canada. ✉email: yang.teoh@mail.utoronto.ca

Prosocial behaviour requires people to balance their own interests against others' welfare. Faced with such conflicts, people sometimes altruistically sacrifice self-interest to help others[1–5], but debates about why and how have preoccupied philosophers, economists and psychologists for centuries[6,7]. Recent work has tried to explain altruistic choices using dual-process models, which assume these choices result from competing dispositional preferences that evolve over time[8,9]. This suggests that one of the keys to enhancing prosociality may lie in understanding whether prosocial dispositions derive from intuition (which arises rapidly and automatically) or controlled reflection (which emerges slowly and only under optimal processing conditions). Resolving this debate would identify fundamental psychological processes that sustain social life, as well as practical approaches to increase prosocial behaviour.

Unfortunately, work on this question has led to conflicting results and conclusions. Using response times as a proxy for automaticity and control has suggested both that prosociality may be rapid and intuitive[8–10] and that it requires lengthy deliberation[11–14]. Although recent work has called into question inferences drawn from deliberation times[15–17], stronger causal manipulations that attempt to interfere with controlled processing using time pressure or instructions to respond intuitively have also led to conflicting conclusions, with people sometimes becoming more selfish[18,19] and sometimes becoming more generous[8,9,20–22]. More recently, some researchers have proposed that some individuals have intuitively generous dispositions, while others are more intuitively selfish[15,21,23]. Yet other work suggests that changes in choices may not necessarily reflect differences in preferences, but rather differences in choice precision[24–26]. Thus a crucial set of questions remains unanswered despite more than a decade of work: when choosing to act altruistically, does generosity or selfishness come first, and if so, why[15,21–23,26]?

Here, we propose a simple mechanistic alternative to dual-process models: when making choices involving conflict between self and others, decision-makers are not simply driven by the fixed and sequential activation of intuitive and controlled processes. Instead, they use a single process—the serial, prioritised deployment of attention—to actively and strategically gather information about their own and others' outcomes according to their dispositional social preferences. Such a model is consistent with research showing that selfish individuals attend more to self-relevant information while prosocial individuals attend more to other-relevant information[27], but goes a step further in suggesting that time pressure (which may force individuals to make fast rather than fully informed choices) should amplify the strategic deployment of attention towards information prioritised by the individual[28,29]. For example, a selfish individual with little time to decide should first ascertain a choice's consequences for herself, and only shift to processing its consequences for others if sufficient time remains. For someone who cares more about others, the order of priorities should be reversed. This strategic deployment of attention could thus lead to changes in choice under time pressure, since research suggests that attended information receives more weight during evaluation[30–35]. While these dynamics might produce patterns that appear on surface to support dual-process models of choice, they would actually stem from this fundamentally different set of mechanisms.

Our model makes three empirical predictions. First, time pressure should result not only in systematic changes in prosociality, but also systematic shifts in attention-allocation towards self- and other-relevant information. Second, these attentional changes should reflect individual differences in how much an individual cares about others' welfare relative to self-interest. Finally, these attentional shifts, combined with time pressure's constraints on serial processing, should better account for changes in generosity under time pressure than intuition-driven changes in social preferences.

To test these hypotheses, we experimentally constrained processing using time pressure and measured the influence of this manipulation on participants' generosity and eye-gaze within a modified dictator game paradigm. We found that time pressure neither consistently increased nor decreased generosity. Rather, supporting our hypotheses, individuals appeared to strategically deploy their attention under time pressure, which in turn predicted changes in generosity. Determining whether changes in gaze were driven by underlying preferences (which might represent a core individual difference present under both time pressure and free response conditions) requires some way to measure those preferences. However, if social preferences drive attention and attention drives the very choices used to infer preferences, this leads to a circularity of inference that makes causal analysis difficult. Thus, to identify underlying preferences independent of attention's effect on choice, and determine how those preferences might shape eye-gaze, we developed an extension of attentional drift-diffusion models (ADDMs)[30,34,35] to simultaneously incorporate and account for real-time eye-movements during choice. As expected, our computational model showed that individuals exhibit stable social preferences that predicted generosity across both conditions. Importantly, we found that the model predicted both early attentional biases and how these biases changed under time pressure. Further confirming our hypotheses, individual differences in attention interacted with time pressure and dispositional social preferences to predict changes in generosity, while potential markers of intuition-driven preferences did not. Finally, forcing individuals to look at others' outcomes rather than their own increased generosity, but only under time pressure, illustrating both the power and limits of attention's causal influence on choice. Thus, our model suggests that altruistic choice dynamics result from dynamic attentional selection as opposed to the sequential activation of intuitive and controlled processes[8,21].

## Result

**Overview**. To examine how time pressure influenced altruistic choices, we asked participants in Study 1 ($N = 60$) to make decisions about proposals consisting of monetary trade-offs between themselves ($Self) and an anonymous partner ($Other) under high or low time pressure (1.5 s vs. 10 s). Participants were asked to accept or reject proposals on each trial relative to a default payout of $50 each (Fig. 1). Proposals for $Self and $Other varied from $0 to $100 each such that any party's gain relative to the default was accompanied by the other party's loss. To incentivize choices, participants were informed that their choice on one randomly selected trial would determine payoffs for themselves and their partner at the end of the experiment. We defined generous behaviour as accepting (rejecting) a proposal wherein participants' losses (gains) accompanied gains (losses) for their partner. To measure selective attention, we tracked participants' eye-movements as they made their choices, defining rectangular areas-of-interest (AOIs) around the two sources of information (i.e., $Self and $Other). Participants' fixations within these AOIs signalled attentional selection of the respective information sources. To confirm the robustness of the behavioural effects, we also ran two behavioural replication studies ($N_{REP1} = 65$ and $N_{REP2} = 49$, see Supplementary Note 1).

**Time pressure moderates individual differences in generosity.** Dual-process models that assume prosociality requires control predict increases in selfish choice under time pressure.

Conversely, models that assume prosocial intuitions predict increases in generosity. We thus sought to determine whether time pressure led to systematic changes in the proportion of generous and selfish choices. As expected, individuals made faster choices under high time pressure ($M_{high} = 0.860$ s, $M_{low} = 3.710$ s, $SE_{diff} = 0.128$ s, $t_{59} = -22.333$, $p < 0.001$). Time pressure also resulted in a small but significant decrease in the proportion of generous choices ($M_{high} = 0.365$, $M_{low} = 0.411$, $SE_{diff} = 0.011$, $t_{59} = -4.167$, $p < 0.001$, Fig. 2a). However, this effect failed to replicate in either replication study (see Supplementary Note 1), contradicting dual-process models that assume that all individuals have the same set of automatic preferences (i.e., either all prosocial or all selfish).

We next explored whether time pressure's effects vary systematically across individuals[15,17,21], testing two possibilities. On one hand, time pressure could simply exacerbate the same tendencies present in free responses. In this case, relative generosity or selfishness in the low time pressure condition should become more extreme under time pressure. On the other hand, consistent with our serial attention hypothesis, time

pressure might produce or unmask biases that are mitigated with extra time. In this case, extreme generosity or selfishness under high time pressure should generally subside with time. Consistent with this latter idea, individuals who made more selfish choices under high time pressure became less selfish with more time whereas individuals who made more generous choices under high time pressure became less generous (Pearson's $r = -0.313$, $t_{58} = -2.513$, $p < 0.05$, Fig. 2b). However, individuals' generosity under low time pressure was not associated with time pressures' effects on generosity (Pearson's $r = -0.101$, $t_{58} = -0.776$, $p = 0.44$, Fig. 2c). Importantly, these effects were robust in replication samples (see Supplementary Note 1) and were not explained by alternative accounts such as regression to the mean (Supplementary Note 2). This result contrasts with work suggesting that time pressure exacerbates individual differences observed under naturalistic response conditions[21]. Rather, our results suggest an additional source of between-individual variance emerges only under time pressure and is mitigated when people had more time to decide. We hypothesised that these biases reflect changes in early attention's influence on choice under time pressure and therefore sought to understand the mechanisms underlying these changes.

**Early attentional priorities emerge under time pressure.** Our model emphasises the importance of attentional priorities in determining choice-outcomes, suggesting two possibilities for time pressure's effects. First, time pressure might simply enhance the influence of existing early attentional biases on choices. Alternatively, time pressure might induce early attentional biases towards social priorities to cope with processing constraints. To test whether such priorities exist by default, or emerge under time pressure, we analysed millisecond-by-millisecond eye-movements during choice, classifying gaze-position of each participant ($N = 57$) at each moment as falling in the Self AOI, Other AOI, or neither. Under high time pressure, we found that participants as a group exhibited a significant gaze-bias towards their own outcomes during the first 294 ms of the trial ($ps < 0.01$, permutation-corrected, see Methods and Fig. 3a). Under low time pressure, participants exhibited a non-significant bias (Fig. 3b). A direct comparison showed that biases towards self-information under high time pressure were significantly larger during the first 286 ms of the trial compared to those under low time pressure ($ps < 0.01$, permutation-corrected), suggesting that time pressure led to an emergence of early selfish gaze-biases. This comparison also revealed a period of more other-oriented gaze under high time pressure from 353 to 575 ms, and a second, later period of greater self-oriented gaze under high time pressure, 718–837 ms ($ps <$

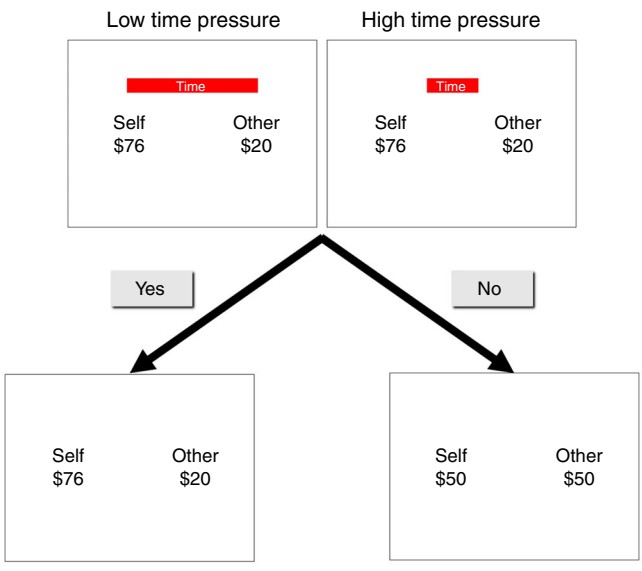

**Fig. 1 Trial structure.** Subjects saw a proposed monetary allocation for Self and Other under time pressure or not (indicated by a red bar) and chose whether to accept or reject the proposal. If they accepted, the proposal was implemented as shown. If they rejected, both they and their partner received a default of $50 each.

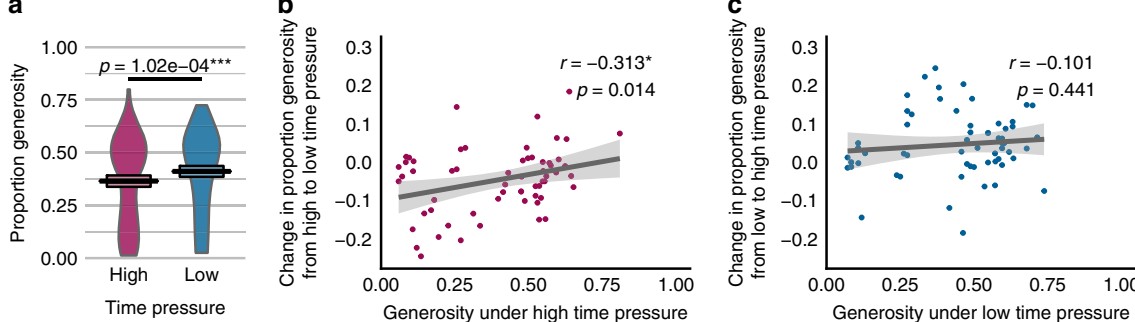

**Fig. 2 Time pressure's effect on generosity in Study 1.** Mean differences across time pressure conditions are displayed in **a**. Violin plots illustrate the distribution of participants' means, coloured by time pressure condition. Central solid line indicates group mean with upper and lower bounds of the box plot indicating the standard error. Prediction of changes in generosity as a function of proportion generous choices under high time pressure **b** and as a function of generosity under low time pressure **c**. Each point represents a single subject ($N = 60$). *$p < 0.05$, **$p < 0.01$, ***$p < 0.001$ for two-tailed **a** paired $t$ test, **b**, **c** one sample $t$ test. Source data are provided as a Source Data file.

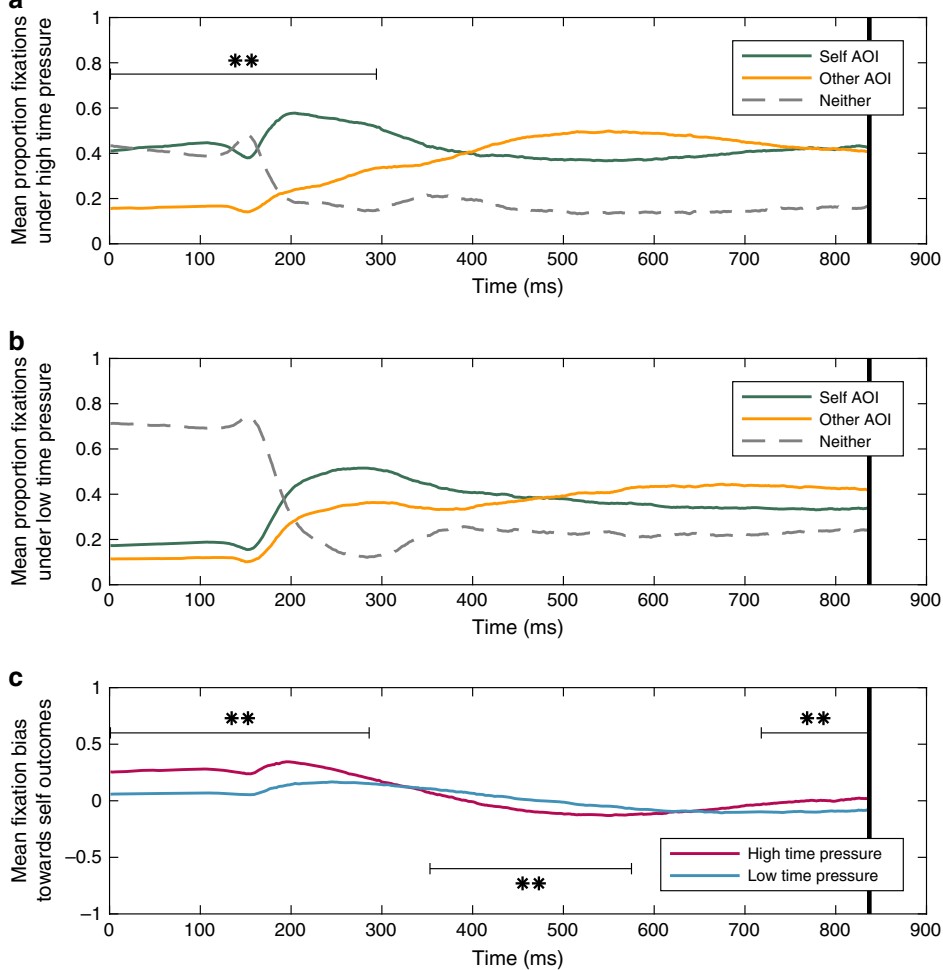

**Fig. 3 Attention dynamics of altruistic choice across time.** Millisecond-to-millisecond proportion of gaze directed to Self AOI, Other AOI or neither under **a** high time pressure, and **b** low time pressure. **c** The difference in proportion of trials with gaze position in Self AOI and Other AOI under high and low time pressure. Lines are coloured by AOI. Bars with asterisks (**) denote time periods (clusters) of permutation-corrected significance (cluster $t$ greater than 95% of simulated $t_{max}$) at a threshold of two-tailed $p < 0.01$ for $z$-tests of the logistic mixed-effects model parameters: **a**, **b** intercept and **c** main effect of time pressure. Comparisons were conducted up until 837 ms, after which >50% of trials terminated under the high time pressure condition. Source data are provided as a Source Data file.

0.01, permutation-corrected), although these biases were not significant in either condition separately (Fig. 3c).

Although we observed an overall selfish gaze-bias under time pressure, we also predicted variability in this effect across participants since individuals should cope with time pressure by prioritising information according to their social preferences but show less consistent gaze behaviour under low time pressure, when such prioritisation is not required. Consistent with this hypothesis, Brown–Forsythe tests of equal variances[36] showed that gaze-biases were significantly more variable (i.e., displayed more extreme values) under time pressure ($F_{1,108} = 4.157$, $p < 0.05$). Furthermore, stronger selfish gaze-biases under high time pressure predicted a lessening of these gaze-biases (i.e., more other-oriented gaze) when given more time, while the reverse (i.e., more self-oriented gaze) held for individuals who were biased towards other-information under time pressure (Pearson's $r = -0.548$, $t_{55} = -4.856$, $p < 0.001$). However, gaze-biases under low time pressure were not associated with changes in gaze under high time pressure (Pearson's $r = 0.039$, $t_{55} = 0.286$, $p = 0.78$). In other words, early attentional biases towards social priorities (self- and other-information) emerged under high time pressure but were attenuated when individuals were given more time to choose.

**Early gaze predicts choice.** If attention filters and selects information in a way that causally influences choice evaluation, then attentional biases should predict individual differences in generosity. Furthermore, limits on extended processing under time pressure should enhance the effect of these biases. To test these hypotheses, we conducted a logistic mixed-effects regression predicting proportion generosity as a function of early gaze, time pressure and their interaction, controlling for later gaze (see Supplementary Table 1). As expected, we found a main effect of time pressure on generosity ($b = -0.0706$, SE $= 0.0268$, $z = -2.637$, $p < 0.01$, $R^2 = 0.008$). However, we also found that self-biased early gaze negatively predicted generosity ($b = -0.494$, SE $= 0.169$, $z = -2.930$, $p < 0.01$, $R^2 = 0.108$), especially under time pressure (interaction: $b = -0.151$, SE $= 0.072$, $z = -2.103$, $p < 0.05$, $R^2 = 0.011$, Fig. 4). Results were similar if first-fixation position rather than proportion early gaze was used as a measure of attention-allocation (see Supplementary Note 3). Together with the results above, these findings suggest that time pressure's effects on generosity are partially driven by its interaction with early gaze.

**The gaze-informed attentional drift-diffusion model.** Although the foregoing results are suggestive, they have yet to demonstrate

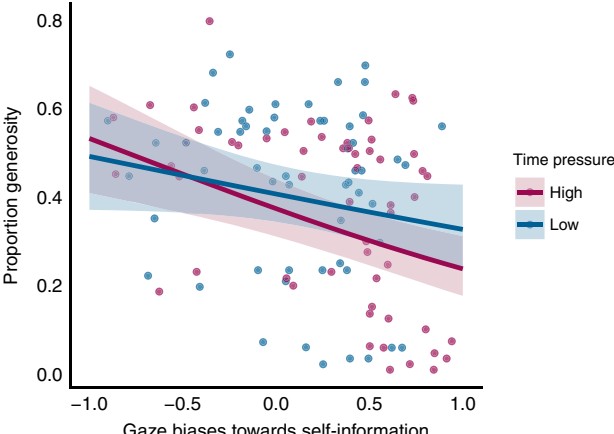

**Fig. 4 Time pressure moderates gaze-biases' effects on generosity.** Each point represents a single subject ($N = 57$), coloured by time pressure condition. Solid lines represent the predicted group averages extracted from the general linear model. Shaded regions represent the 95% confidence interval of the predicted values centred at the predicted group mean, coloured by time pressure condition. Source data are provided as a Source Data file.

two critical assumptions of our model: that changes in attention causally influence choice, and that those changes reflect underlying preferences for certain kinds of information over others. However, if underlying preferences shape attention, but attention also influences the choices used to infer underlying preferences, this circularity can make it challenging to detect and distinguish the independent influence of the two. Thus, we sought to isolate attention's influence, and estimate social preferences independently of this influence, by explicitly incorporating it into a model of the choice process. This allowed us to determine if changes in generosity under time pressure were driven by individual differences in attention, intuitive preferences, or both.

To do this, we developed an eight-parameter gaze-informed multi-attribute extension of the attentional drift-diffusion model (ADDM)[30,35] (see "Methods"). Our model makes several key assumptions. First, when choosing to act selfishly or generously, we assume that people construct evidence for or against the available options based on a weighted sum of both self-interest and social preferences about the other and fairness[37]. These weights (i.e., $w_{self}$, $w_{other}$, $w_{fairness}$) parameterise individuals' overall social preferences as they accumulate evidence towards a threshold for choice[38]. This threshold is determined by an initial height parameter $b$, and its exponential decay, $d$, towards 0 over time within the trial[24,26,39]. Second, we assume that visual attention towards one type of information (i.e., $Self or $Other) momentarily reduces the influence of unattended information in the accumulation process by a factor $\theta$. Unlike previous applications of the ADDM, which model attention generically using group or trial averages[30,31,35], our model makes full use of each individual's moment-by-moment gaze position to determine exactly when different attributes (1) enter consideration[32] and (2) receive amplification. Finally, the model also allows people to have starting biases towards a particular response (i.e., to choose the proposal or default, stbias, or to choose generously or selfishly, genbias) even before the value of a specific stimulus is known[21].

Our model formalises the intuitive notion implicit in dual-process models that value-construction of options unfolds over time, and that interrupting the evidence-accumulation process through time pressure can alter choices. It considers three possible mechanisms for these changes. The first mechanism (a

Social Heuristics[8,9] or dual-process account) assumes that early evidence differs from later evidence, because automatic processing generates evidence more rapidly than controlled processing[40]. Thus, forcing people to choose quickly pre-empts late-emerging evidence, revealing the contents of intuitive preferences[8,13]. This predicts that changes in weight parameters should predict changes in generosity since intuitively selfish individuals initially weight self-interest more highly while intuitively prosocial individuals initially weight others' outcomes more highly. The second mechanism (our prioritised attention model) also hypothesises differences in early vs. later evidence, but posits that this results not from fixed activation of internal processes, but the dynamic and strategic allocation of attention. This predicts that changes in attention, rather than changes in weight parameters, should relate to changes in generosity. Finally, a third mechanism (the biased DDM[21]) suggests that time pressure exacerbates the effect of starting biases whose influence on choice is larger when less evidence is considered, resulting in automatic default responses towards generosity or selfishness[21]. This predicts that the estimated parameter values for starting biases should predict changes in generosity under time pressure.

**Absolute model fit.** Before examining model parameters, we verified that our model fit the data well and captured both inter-individual (Fig. 5a, b) and intra-individual (Fig. 5c, d) variability. We observed strong associations between simulated and observed changes across time pressure conditions in generosity (Pearson's $r = 0.784$, $t_{48} = 8.738$, $p < 0.001$) and log-transformed RTs (Pearson's $r = 0.970$, $t_{48} = 27.793$, $p < 0.001$). In addition, the model captured trial-by-trial differences in acceptance rates and RTs within individuals (acceptance rate: high time pressure: mean Pearson's $r = 0.875$, SE $= 0.0246$, $t_{49} = 35.501$, $p < 0.001$; low time pressure: mean Pearson's $r = 0.911$, SE $= 0.0092$, $t_{49} = 99.154$, $p < 0.001$; logRT: high time pressure: mean Pearson's $r = 0.616$, SE $= 0.0299$, $t_{49} = 20.609$, $p < 0.001$; low time pressure: mean Pearson's $r = 0.396$, SE $= 0.0434$, $t_{49} = 9.114$, $p < 0.001$, see Methods). Model-extracted parameters quantifying dispositional social preferences were also predictive of generosity under high time pressure ($w_{self}$: $b = -41.120$, SE $= 3.837$, $z = -10.003$, $p < 0.001$, $R^2 = 0.276$; $w_{other}$: $b = 74.418$, SE $= 3.837$, $z = 19.393$, $p < 0.001$, $R^2 = 0.656$; $w_{fair}$: $b = 32.008$, SE $= 8.322$, $z = 3.846$, $p < 0.001$, $R^2 = 0.062$) and low time pressure ($w_{self}$: $b = -27.239$, SE $= 3.794$, $z = -7.179$, $p < 0.001$, $R^2 = 0.120$; $w_{other}$: $b = 60.077$, SE $= 3.507$, $z = 17.129$, $p < 0.001$, $R^2 = 0.536$; $w_{fair}$: $b = 29.469$, SE $= 7.921$, $z = 3.720$, $p < 0.001$, $R^2 = 0.044$).

**Model comparison.** To assess the added explanatory and predictive value of accounting for attentional dynamics in our gaze-informed ADDM, we first fit a version of Chen and Krajbich's biased DDM[21] assuming no influence of attention. Using split-half cross-validation, we compared the two models on their out-of-sample predictive accuracy, both for change across time pressure conditions, and average generosity in each condition. Notably, the biased DDM failed to predict changes due to time pressure (Pearson's $r = 0.219$, $t_{48} = 1.558$, $p = 0.126$), while the gaze-informed ADDM was significantly more accurate (Pearson's $r = 0.483$, $t_{48} = 3.818$, $p < 0.001$; comparison: Fisher's $t = 2.16$, one-tailed $p = 0.02$, see Fig. 6). We observed similar differences when examining predictive accuracy for generosity in each condition separately. While both models accurately predicted generosity under high time pressure (gaze-informed ADDM: Pearson's $r = 0.911$, $t_{48} = 15.324$, $p < 0.001$; Biased DDM: Pearson's $r = 0.783$, $t_{48} = 8.730$, $p < 0.001$) and low time pressure (gaze-informed ADDM: Pearson's $r = 0.859$, $t_{48} = 11.630$, $p < 0.001$; biased DDM: Pearson's $r = 0.702$, $t_{48} = 6.825$,

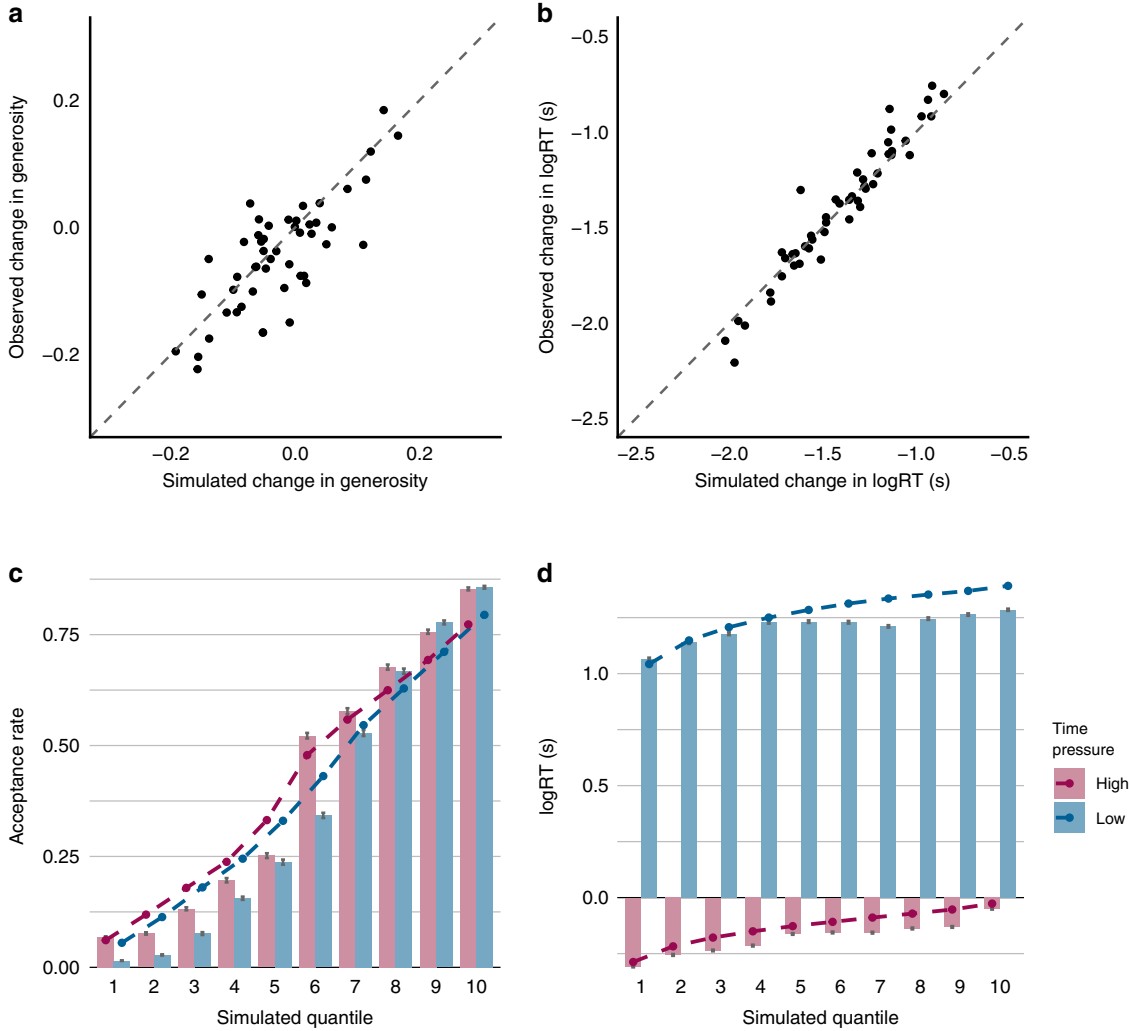

**Fig. 5 Simulated vs. observed behaviour in the gaze-informed ADDM.** Model versus observed inter-individual differences in **a** change in proportion generosity under time pressure, and **b** change in logRT under time pressure. Each point represents a subject ($N = 50$), and points on the dashed line indicate perfect simulation of the observed data. Bottom panels illustrate simulated versus observed intra-individual differences, coloured by time pressure condition, in **c** acceptance of proposed offers, and **d** logRT, binned into ten model-predicted quantiles within individuals. Column height represents the observed group averages with error bars denoting one standard error above and below the mean. Points on the dashed line represent the simulated averages within these quantiles across individual subjects ($N = 50$). Source data are provided as a Source Data file.

$p < 0.001$), the gaze-informed ADDM was significantly more accurate in both cases (high time pressure: Fisher's $t = 3.58$, one-tailed $p < 0.001$; low time pressure: Fisher's $t = 3.44$, one-tailed $p < 0.001$). Thus, while social response biases may partially drive choice behaviour, they cannot fully account for time pressure's effects on altruistic choice. Instead, accounting for attentional dynamics seems necessary to fully capture time pressure's effects on altruistic choice.

Given our strong hypotheses that attention drives time pressure's effects on choice, we also expected exclusion of the social bias parameter from our model to minimally influence predictive accuracy. As expected, a more parsimonious version of the gaze-informed model that excluded social response biases was just as accurate as the full model in predicting changes in generosity (nested model: Pearson's $r = 0.500$, $t_{48} = 3.998$, $p < 0.001$, Fisher's $t = -0.29$, two-tailed $p = 0.77$), and generosity under low time pressure (nested model: Pearson's $r = 0.848$, $t_{48} = 11.091$, $p < 0.001$, Fisher's $t = 0.74$, two-tailed $p = .46$). However, while the more parsimonious model accurately predicted generosity under high time pressure (nested model: Pearson's $r = 0.883$, $t_{48} = 13.014$, $p < 0.001$), the full model was

slightly but significantly more accurate (Fisher's $t = 2.14$, two-tailed $p = 0.04$). Thus, while social response biases may contribute to choice behaviour under time pressure, a more parsimonious model accounting only for attentional dynamics is sufficient to capture time pressure's effects on change in altruistic choice.

Additional model simulations further suggest that if attention drives early values, models that are ignorant of attention could erroneously attribute rapid selfish or generous responses to response biases, possibly explaining previous work attributing time pressure's effects to such biases (see Supplementary Note 4). Altogether, these results strongly suggest that the temporal dynamics of attention-control play a critical role in driving the effects of time pressure on altruistic choice.

**Time pressure's effect on computational parameters.** Having shown that our model fit the data well, and that attentional dynamics are both necessary and sufficient for explaining time pressure's effects on altruistic choice, we extracted for further analyses the estimated parameters from the full gaze-informed ADDM fitted to all of the observed data. We then computed

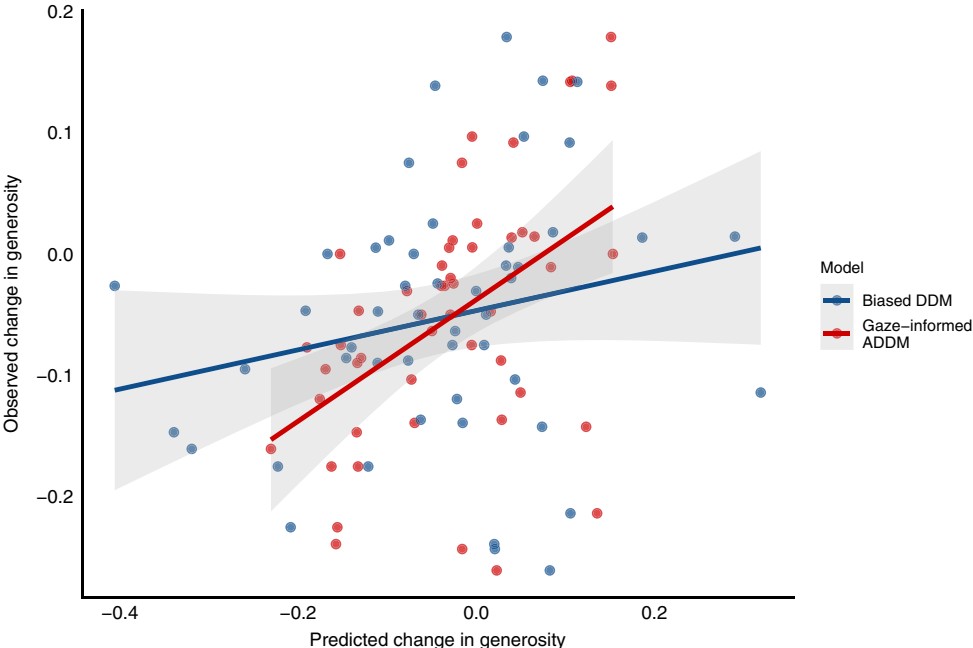

**Fig. 6 Model comparison.** Comparison between the gaze-informed ADDM and the biased DDM in their predictive accuracy of changes in generosity under time pressure. Each point represents a subject (N = 50), and points on the dashed line indicate perfect prediction of the observed data, coloured by model. Source data are provided as a Source Data file.

paired $t$ tests of changes in each parameter under time pressure, Bonferroni-corrected (eight comparisons, see Supplementary Table 2). As expected, time pressure dramatically decreased decision thresholds, $b$, ($p < 0.001$, corrected) and increased their decay rate, $d$ ($p < 0.001$, corrected). Comparatively, changes in response-option starting biases, stbias ($p = 0.018$, uncorrected) and attentional discount, $\theta$ ($p = 0.032$, uncorrected) did not survive correction. We also observed no significant changes in intuitive social response biases, genbias ($p = 0.24$, uncorrected), although people were selfishly biased on average ($M_{\mathrm{average}} = -0.0377$, SE $= 0.0118$, $t_{49} = -3.173$, $p < 0.01$).

We next turned our attention to weights on self, other, and fairness, since changes in these parameters come closest to capturing the idea that rapid or intuitive preferences differ from more deliberative ones. Specifically, dual-process models suggest that time pressure should increase the weighting of more rapidly processed attributes, driving changes in generosity[40,41]. Comparisons revealed an increase in $w_{\mathrm{self}}$ ($p < 0.05$, corrected) and $w_{\mathrm{fairness}}$ ($p < 0.01$, corrected) under time pressure, but little change in $w_{\mathrm{other}}$ ($p = 0.59$, uncorrected). These results suggest that time pressure primarily reduces the decision threshold, but may also increase preferential processing of self-interest and inequality. We return to this latter point below in analyses determining whether attribute weights or gaze-biases better explain changes in generosity under time pressure.

**Stable social preferences shape early attentional dynamics.** Our attention model predicts that people's dispositional social preferences (i.e., the weight they place on self vs. others) should shape how they shift their attention under time pressure. By modelling attention, we were able to obtain a cleaner measure of social preferences to test this important prediction. Importantly, linear mixed-effects regression predicting early gaze-biases from model parameters (N = 50) revealed that concern for other's outcomes, average $w_{\mathrm{other}}$, was negatively associated with selfish gaze-biases ($b = -12.819$, SE $= 5.294$, $t_{47} = -2.421$, $p < 0.05$, semi-partial $R^2 = 0.101$), especially under time pressure

(interaction: $b = -3.656$, SE $= 1.568$, $t_{47} = -2.332$, $p < 0.05$, semi-partial $R^2 = 0.009$). Early gaze-biases were not associated with individuals' weight on self-interest, $w_{\mathrm{self}}$, regardless of time pressure condition ($ps > 0.05$, see Supplementary Table 3). Furthermore, individuals' average $w_{\mathrm{other}}$ also negatively predicted tendencies to shift gaze priorities towards \$Self under time pressure ($b = -7.312$, SE $= 3.136$, $t_{47} = -2.332$, $p < 0.05$, semi-partial $R^2 = 0.104$), even when controlling for average $w_{\mathrm{self}}$ ($b = 5.273$, SE $= 3.907$, $t_{47} = 1.350$, $p = 0.18$, semi-partial $R^2 = 0.037$). Together these results support our model of prioritised attention: individuals' early attention tracks their dispositional social preferences, especially under time pressure.

**Attentional shifts drive changes in generosity.** Having observed that time pressure may alter both the weights assigned to self-interest and other-focused concerns, and the attention paid to them, we sought to determine how such alterations, as well as default social response biases[21], influence changes in generosity. We thus conducted stepwise regression on changes in generosity under time pressure (N = 50, see Supplementary Table 4). The most parsimonious model revealed only a significant main effect of average $w_{\mathrm{other}}$ ($b = -6.077$, SE $= 2.881$, $t_{38} = 2.109$, $p < 0.05$, semi-partial $R^2 = 0.105$), and three-way interaction between early gaze-biases, their change under time pressure, and average $w_{\mathrm{other}}$ ($b = 33.228$, SE $= 14.773$, $t_{38} = -2.249$, $p < 0.05$, semi-partial $R^2 = 0.117$). We observed no effect of changes in weights or default social biases.

In other words, for people who cared about others' outcomes (i.e., +1SD in $w_{\mathrm{other}}$), inattention to those outcomes was detrimental to generosity under time pressure: becoming more self-focused in their early gaze tended to reduce generosity under time pressure, given that average early gaze was not already highly self-biased (two-way interaction: $b = 0.385$, SE $= 0.195$, $t_{38} = 1.975$, $p = 0.056$; simple effect of change in early gaze-biases at $-1$ SD early gaze-bias: $b = -0.231$, SE $= 0.128$, $t_{38} = -1.802$, $p = 0.080$). When early gaze-biases were already highly self-biased (mean or +1 SD), further biasing of early gaze towards self-

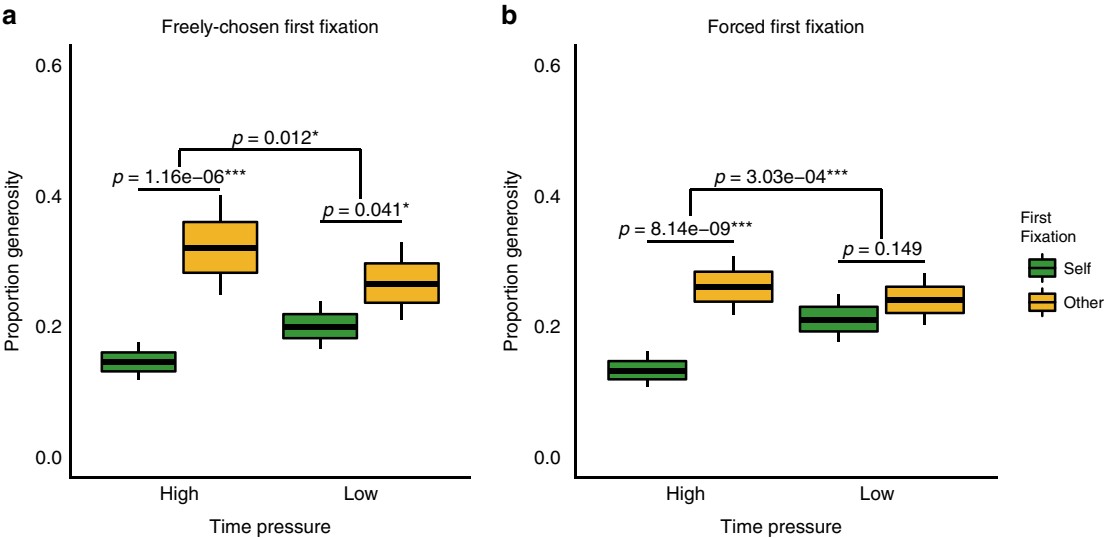

**Fig. 7 First fixation predicts generosity under time pressure but not time delay.** Effects of first fixation on generosity moderated by time pressure in **a** free attention trials ($N = 200$ subjects) and **b** forced attention trials ($N = 200$ subjects). Central line in boxplots indicate estimated means and upper and lower bounds indicate one standard error above and below the mean, coloured by first-fixation position. The whiskers indicate the 95% confidence interval for the estimated mean. *$p < 0.05$, **$p < 0.01$, ***$p < 0.001$ for two-tailed $z$-tests of the regression estimate. Source data are provided as a Source Data file.

outcomes did not change generosity (simple effect at mean: $b = -0.056$, SE $= 0.071$, $t_{38} = -0.788$, $p = 0.436$; at +1 SD: $b = 0.120$, SE $= 0.096$, $t_{38} = 1.245$, $p = 0.221$). For those who placed little weight on others' outcomes (i.e., mean or −1 SD in $w_{other}$), early attention did not matter (simple effects of average and change in early gaze and their two-way interaction, $ps > 0.05$) Thus, early attentional biases robustly drive changes in generosity under time pressure and provide a parsimonious account of choice dynamics.

**Forced attention biases generosity under time pressure.** Our model thus far suggests a causal account wherein attention, driven by social preferences, influences the choices people make under time pressure. However, to directly test attention's causal influence on altruistic choice, we conducted a second, on-line study where we manipulated participant's attention to $Self and $Other. In this study, participants ($N = 200$) completed similar dictator games under high and low time pressure (see "Methods"), using mouse clicks to reveal choice attributes ($Self or $Other). Importantly, on some trials, they could choose which attribute to click on first. On other trials, they were forced to click on either $Self or $Other first. For parallelism with Study 1, we use the term fixation to describe when participants clicked on an attribute. If early attention derives its influence on choice only through its relationship with dispositional social preferences, first-fixations should have little to no effect when they are exogenously controlled. However, if attention has a direct causal influence on choice, then it should influence behaviour even when determined by outside forces.

Replicating Study 1, participants in Study 2 were more likely to choose selfishly if they freely looked first at $Self, specifically under time pressure (interaction: $b = 0.648$, SE $= 0.259$, $z = 2.504$, $p = 0.012$; simple effect of first-fixation under high time pressure: $b = -1.016$, SE $= 0.209$, $z = -4.855$, $p < 0.001$; low time pressure: $b = -0.368$, SE $= 0.180$, $z = -2.048$, $p = 0.041$, Fig. 7a). Moreover, generosity in forced-attention trials (which provides a measure of dispositional social preferences controlling for attention) predicted how selfishly oriented participants' freely chosen first-fixations were, especially under time pressure (interaction: $b = 0.710$, SE $= 0.258$, $z = 2.748$, $p = 0.006$; simple effect of generosity under high time pressure: $b = -2.636$, SE $=

0.559$, $z = -4.718$, $p < 0.001$; low time pressure: $b = -1.926$, SE $= 0.557$, $z = -3.461$, $p < 0.001$). These findings corroborate results from Study 1 suggesting that dispositional social preferences drive attentional priorities, particularly under time pressure, and that these attentional priorities predict choice.

Finally, we confirmed a causal effect of attention on choice independent of social preferences: generosity was strongly influenced by whether participants were forced to fixate on $Self or $Other first, but only under time pressure (interaction: $b = 0.664$, SE $= 0.184$, $z = 3.601$, $p < 0.001$; simple effect of first-fixation under high time pressure: $b = -0.833$, SE $= 0.144$, $z = -5.766$, $p < 0.001$; low time pressure: $b = -0.169$, SE $= 0.117$, $z = -1.444$, $p = 0.149$, Fig. 7b). Thus, even when holding dispositional social preferences constant, directing participants' attention towards their own or others' outcomes made them more selfish or more generous, respectively.

## Discussion
Do preferences for acting generously evolve over time? Our results suggest they do but point to a markedly different explanatory mechanism than extant dual-process models: the serial, prioritised deployment of attention to preferred information during information-search. Three pieces of evidence support this interpretation. First, we found that individuals differ systematically in the extent to which they focused first on self- or other-relevant information under time pressure. Second, a sophisticated, gaze-informed computational model of choice suggested that underlying social preferences drive these initial attentional biases, particularly under time pressure. Third, these attentional biases causally predicted changes in generosity under time pressure. Our findings help to make sense of the growing body of literature suggesting that time pressure neither consistently increases nor decreases generosity[8,19,22], but instead reveals individual differences in social preferences[21,23]. Moreover, while some of our findings hint at the possibility of fast and slow systems of processing[8,13,21,42,43], they point firmly to the importance of attentional priorities in the face of processing constraints.

Our results suggest an important insight into human prosociality: individuals tend to prioritise information about self-interest over information about others' welfare. They

demonstrated a tendency to look first at their own outcomes, particularly under time pressure, and the model-estimated influence of this information on choice, even after accounting for attention, was significantly higher. Yet these results appear to contradict the idea that individuals often default to pro-social responses (social heuristic hypothesis)[8,15,44]. What might reconcile such findings? Two possibilities are immediately apparent.

First, much of the research demonstrating intuitive biases uses single-shot games in which participants self-generate monetary splits, whereas our study used repeated presentation of specific, experimenter-determined payouts. Perhaps response-modality, or the repeated exposure to games of this sort[9] results in different decision processes. However, in supplementary analyses, we incorporated a series of one-shot games into our measure of prosociality and found that generosity in one-shot and repeated binary choices were strongly correlated, suggesting that they both tap into a common set of processes (see Supplementary Note 5).

Instead, we think a second explanation is more likely. Much of the literature demonstrating default pro-social biases has emerged during the study of strategic cooperative behaviour, such as the Ultimatum or Public Goods Game, rather than purely altruistic behaviour[8,9,15,17]. We suspect that when one's own outcomes are more clearly predicated on the behaviour or outcomes of others (as in the Ultimatum Game), attention shifts more decisively toward social information, yielding pro-social biases under time pressure. Future work using eye-tracking could test such a hypothesis by comparing the pattern of eye-gaze during Dictator and Ultimatum games.

Examining whether people flexibly direct attention towards self-relevant information in a context-sensitive manner may also unveil the mechanism that drives the emergence of attentional biases under time pressure. Based on the relationship between model-identified underlying preferences and changes in attentional bias, we speculate that strategic goal-directed mechanisms are at play; people might be able to look in anticipation toward self-relevant information given salient cues about where it appears. However, it is also possible that components of these early attentional biases are intuitive expressions of social preferences[27,45], driven by well-established characteristics of the automatic attention system[46,47]. Such an interpretation would be consistent with studies showing that previously rewarding stimuli capture attention automatically, regardless of the current goal context[48]. In our paradigm, the location of self-relevant information was constant throughout. Thus, to the extent that selfish individuals disproportionately value self-outcomes, the screen location signalling this value might come to draw attention in an automatic and habitual fashion. Future work manipulating the location of self- and other-relevant information from trial to trial in a predictable way might help to determine the relative automaticity or strategic goal-directedness of attentional prioritisation. However, effects of strategic deployment and reward-based attention-capture need not be mutually exclusive. Careful consideration of how these distinct processes support and dynamically shape the twin tasks of attention-allocation and choice-comparison may help to resolve important inconsistencies in the literature.

Furthermore, the exact mechanisms of attention's influence on the evidence-accumulation process remains unclear. Some work suggests that attention supports the initial construction of the choice-set while others posit that attention amplifies the value of evidence in real-time. While we have included both these possibilities as attentional mechanisms in our computational model, the field continues to debate the relative contributions of these mechanisms during choice[32,35,49]. Future work should seek to characterise and disambiguate between these mechanisms and their downstream effects on choice. We suspect that neurocomputational models like the one we have developed here will be key for resolving such issues. Importantly, these general cognitive mechanisms (attention and its multiple interactions with valuation) likely extend beyond altruistic and cooperative decision-making to other domains such as risky decision-making[50] and dietary choices[51,52]. Our model clearly reveals the importance of considering dynamic, reciprocal connections between externally directed attention and valuation. Future work will need to explore how other dynamic processes, including internally directed attention[53,54], memory[55,56] and affective responding[57] shape and interact with value-construction during social behaviour and beyond.

## Methods

**Study 1**. Participants ($N = 60$) completed 160 trials of the modified dictator game (Fig. 2). To manipulate time pressure, participants had to make these choices either within 1.5 s after trial onset (high time pressure, 80 trials) or within 10 s (low time pressure, 80 trials). In the low time pressure condition, participants were also probabilistically notified they had responded fairly quickly and reminded to take their time to make the best choice if they responded before 2 s after trial onset. Supplementary analyses suggest that these prompts to delay responding did appear to encourage more extensive deliberation before choice (see Supplementary Note 6). Participants alternately encountered the low time pressure condition before high time pressure trials in blocks of 20 trials. Additional analyses revealed consistent patterns of effects across all blocks and no effects of block ordering (see Supplementary Note 7).

Stimuli were presented and responses collected using MATLAB and the Psychophysics Toolbox[58,59] on a 23-in. display monitor (100-Hz refresh rate, resolution 1920 × 1080 pixels). For eye-tracking purposes, participants sat in front of the computer screen at a distance of approximately 60 cm, with their head in a chin-rest to minimise head movements.

At the end of the study, a random trial was selected from the participant's choice set and the participant and their partner received the monetary outcomes of their choice on that trial. These outcomes were paid to the participant immediately, and to another participant in the study completing a subsequent experimental session. Thus, most participants completed the task once as the decision maker, receiving the outcome of their choice, and once as a recipient of the outcome of a previous participant's randomly implemented choice. Participants discovered their passive participation as a recipient only at the end of the study. Monetary payoffs presented during the study ranged from $0 to $100 and were converted to real payouts using an exchange rate of 5:1. Participants learned that there would be an exchange rate in the instructions prior to the task but were not informed of the precise ratio until the end of the study. All participants provided informed consent in manners approved by the Research Ethics Board at the University of Toronto.

**Study 2**. Study 2 was designed to examine the causal influence of attention on choice, independent of dispositional social preferences. Towards this end, we recruited participants from Amazon mechanical TURK (Final $N = 200$, 196 pre-registered exclusions) and compensated $5 for their time, plus an amount determined by their choices in the study. Participants first completed a series of five one-shot dictator game where they were given varying amounts of money and asked how much they would like their partner to have on a continuous scale (see Supplementary Note 5). In the main task, participants completed 136 trials of a modified online version of the dictator game in Study 1 (see Supplementary Fig. 1) using Inquisit Web. To measure attention analogous to eye-gaze in Study 1, participants had to click on the location of $Self or $Other to reveal it, and could only view one piece of information at a time. To manipulate time pressure, participants had to make these choices either within 3 s after trial onset (high time pressure, 68 trials) or only after 3 s had elapsed but within 10 s (low time pressure, 68 trials). Subjective reports of time pressure on a single-item scale from 1 (Not at all) to 7 (Extremely) ("How rushed did you feel in trials where you had only 3 s to respond?") provided confirmation for a manipulation check on this modified paradigm ($M_{\text{pressure}} = 5.082$, SD = 1.697). In 22 trials of each of the time pressure conditions, participants could choose to click on either their own outcomes or the other person's outcomes first. In another 23 trials, participants were forced to click on their own outcomes (self-outcomes) first. In the last 23 trials, participants were forced to click on the other person's outcomes (other-outcomes) first. Participants always encountered a low time pressure block followed by a high time pressure block in the practice and beginning of the main experiment but all following high and low time pressure blocks were randomly interleaved.

To signal free vs. forced-click trials, a visual border around the attributes cued what information was available for access (i.e., only self-outcomes, only other-outcomes, or both). Upon clicking, the selected information ($Self/$Other) appeared briefly, after which participants were prompted to access the non-selected piece of information. The duration of each attribute exposure were designed to mimic durations from Study 1. Participants were forced to oscillate between these pieces of information until they made a choice, or the time limit had elapsed. We

recorded the number and order of clicks on each trial and defined each instance of information access as a fixation for subsequent analyses.

At the end of the study, a random trial was selected from the participant's choice set and the participant and their partner received the monetary outcomes of their choice on that trial. These outcomes were paid to the participant immediately, and to another participant in the study completing a subsequent experimental session. Thus, most participants completed the task once as the decision maker, receiving the outcome of their choice, and once as a recipient of the outcome of a previous participant's randomly implemented choice. Participants discovered their passive participation as a recipient only at the end of the study. Monetary payoffs presented during the study ranged from $0 to $100 and were converted to real payouts using an exchange rate of 50:1. All participants provided informed consent in manners approved by the Research Ethics Board at the University of Toronto. All details on experimental design and analyses of Study 2 were preregistered on the Open Science Framework (OSF) with the identifier CHWM3 [https://doi.org/10.17605/OSF.IO/CHWM3].

**Generous choices**. We defined generous choices as trials where the participant accepted (rejected) a smaller (larger) amount of money for themselves compared to the default, in order to help their partner receive a larger amount. Choices were defined as selfish otherwise. Proportions of generous choices were calculated as the number of generous choices a participant made over the number of trials in which a response was recorded. Missed response trials (mean percentage of trials: high time pressure: $M_{study1} = 4.67\%$, $M_{REP1} = 3.03\%$, $M_{REP2} = 2.29\%$, $M_{study2} = 1.03\%$; low time pressure: $M_{study1} = 0.33\%$, $M_{REP1} = 0.03\%$, $M_{REP2} = 0.39\%$, $M_{study2} = 0.03\%$) were excluded from choice analyses.

**Eye tracking**. In Study 1, we recorded eye-movements from the right eye (Pupil-CR tracking mode) using an EyeLink 1000 plus Desktop Mount (SR Research, Ontario, Canada) with a sampling rate of 1000 Hz. Before starting the experiment, the eye tracker was calibrated and validated to ensure tracking accuracy. For the calibration, participants fixated nine random dots on the screen; the eye tracker was then adjusted until the average tracking error of the visual angle was less than 0.5°. Following calibration, a nine-point validation phase was performed. The validation procedure measured the difference between the computed fixation position and the fixation position for the target obtained during calibration using the same nine random dots to ensure that the calibration was accurate and eye position errors were acceptable (average gaze-position error: 0.36° ± 0.01). To analyse the eye-movement data, we defined two (300 × 370 pixels) non-overlapping areas of interest (AOIs) around the two attributes ($Self and $Other), positioned at an equal distance from the centre cross, centred at ($x = 320px$, $y = 540px$) and ($x = 1280px$, $y = 540px$) from the left and top of the screen. Attribute positions (left or right) of $Self and $Other were counterbalanced across subjects to mitigate leftward reading biases. Eye-movement data were analysed between proposal onset and the response. We excluded three participants due to technical difficulties with the eye-tracking equipment during data collection, leaving a total of 57 eligible participants for remaining eye-tracking analyses.

We first extracted unfiltered eye-position data from all eligible participants using the Eyelink Data Viewer 3.1 with millisecond precision, classifying eye-gaze position as within the Self AOI, Other AOI or neither. To identify overall effects of time pressure, we then analysed gaze position across participants using univariate logistic mixed-effect models at every time point until 837 ms, the point at which >50% of trials had terminated under the high time pressure condition. The range of median reaction times under time pressure across subjects varied from 528 to 1084 ms with more than 50% of individuals' median reaction times between 727 and 935 ms. To correct for multiple comparisons across the time points, we performed cluster-based permutation testing using the maximum cluster-level mass statistic, $t_{max}$[60,61]. Clusters were defined as time periods with at least two adjacent, significant time points.

For individual difference analyses, we extracted early gaze-biases, measured as the average proportion of the time individual participants spent fixating on the Self vs. Other AOI in the first 286 ms of the trial (the time period of significant self-focused bias identified by the permutation analysis). For 15 of 9120 trials, in which participants responded before 286 ms, we calculated the proportion bias of early gaze with respect to their reaction time. Measures of later gaze-biases were computed as the proportion of time spent in the Self vs. Other AOI for the remaining trial duration (i.e., overall trial duration—286 ms). In these analyses, we included trials where participants failed to response before the time limit, given that these trials still provide information about individual participants' attentional deployment. Excluding missed response trials (230 of 9120) yielded similar effects across all analyses.

In addition to continuous gaze data, we also conducted all analyses using measures based on first fixation position. In these analyses we excluded all fixations that were shorter than 100 ms and removed trials where participants failed to fixate on either of the two AOIs. These analyses revealed similar results to those using continuous measures of gaze, including the effect of attention on generous choice (see Supplementary Note 3).

**The gaze-informed ADDM**. To investigate the effects of early attention on choice, we adapted the ADDM[30], constraining it by millisecond-by-millisecond trial-specific gaze position for each participant, and combined it with a multi-attribute extension of the DDM for altruistic choice[26]. Evidence for accepting the proposal on each trial (relative to the default) accumulated over time as a function of samples of the expected value (V). We defined the momentary expected value V(t) as the weighted sum of three attributes: self-interest ($Self), concern for others ($Other) and inequality (|$Self − $Other |), with weights on $Self and $Other discounted by a factor θ when not the focus of visual attention, as shown in Eqs. (1–3).

$$V(t) = A(t)_{self} \times w_{self} \times \$Self + A(t)_{other} \times w_{other} \times \$Other + w_{fairness} \times |\$Self - \$Other| + \epsilon(t), \quad (1)$$

$$A(t)_{self} = \begin{cases} 1, & \text{if gaze on Self AOI} \\ 1 - \theta, & \text{otherwise.} \end{cases} \quad (2)$$

$$A(t)_{other} = \begin{cases} 1, & \text{if gaze on Other AOI} \\ 1 - \theta, & \text{otherwise.} \end{cases} \quad (3)$$

We further assumed that weights for a given attribute were 0 if the participant had not yet fixated on the information necessary to compute it (e.g., $w_{other} = 0$ and $w_{fairness} = 0$ if the participant had not yet fixated on $Other at any point prior to t). In other words, we assumed ignorance of a particular attribute until visual fixations confirmed acquisition of the necessary information ($Self for $w_{self}$, $Other for $w_{other}$, $Self and $Other for $w_{fairness}$).

Individual samples of the value V(t) accumulate over time, until they hit one of two choice-defining thresholds, determined by two parameters: an initial height parameter b, as well as a collapse-rate parameter d, capturing the exponential decay of the boundaries towards 0 over time within the trial[24,26,39]. We also employed two response bias parameters: a genbias parameter capturing intuitive social response biases predisposing people towards generous or selfish responses[21], and a starting bias parameter stbias capturing a bias to reject or accept the proposal. These parameters summed together, such that the overall bias, z, towards accepting the proposal on any given trial was defined as shown in Eq. (4).

$$z = \begin{cases} \text{stbias} + \text{genbias}, & \text{if } \$Self < \$Other, \\ \text{stbias} - \text{genbias}, & \text{if } \$Self > \$Other. \end{cases} \quad (4)$$

We fit these 8 parameters [wself, wother, wfair, b, d, stbias, genbias, θ] for each of both time pressure conditions simultaneously to quantify the unobservable dynamics of the decision process. Thus, we obtain two values per parameter that capture individual stability (mean) and contextual effects (change), resulting in 16 parameter values.

Additionally, to account for perceptual and motor processing that adds to response time without influencing the decision process itself, we included two separate percept and motor non-decision parameters, fixing each at 0.08 s based on neurobiological evidence about visual[62] and motor processing times[63]. We added a single instance of the motor latency to simulated RTs of the model following termination of evidence at one of the two boundaries for choice. We added one instance of the percept latency to the final RT for the first instance of fixation on each unique piece of information (i.e., once if the participant only fixated on either Self or Other throughout the course of the trial, or twice if participant fixated on both). We further assumed that the value V(t) during this percept period was a function only of the previously available information, discounted by attention as shown in Eq. (5).

$$V(t) = \begin{cases} 0, & \text{if no previous fixation in AOIs,} \\ (1 - \theta) \times w_{self} \times \$Self + \epsilon(t), & \text{if only sampled Self AOI,} \\ (1 - \theta) \times w_{other} \times \$Other + \epsilon(t), & \text{if only sampled Other AOI.} \end{cases} \quad (5)$$

**The multi-attribute DDM**. For comparison to the gaze-informed ADDM, we also fit a static multi-attribute DDM[21]. This model was identical to the gaze-informed model, with the exception that it did not include a discount parameter θ or information about eye fixations, and collapsed motor and percept parameters into a single non-decision parameter, ndt, that was estimated from the data. Thus, the model estimated 16 free parameters: 8 parameters related to average $w_{self}$, $w_{other}$, $w_{fairness}$, b, d, stbias, genbias, and ndt and 8 related to change in these parameters due to time pressure.

**Model estimation**. To examine overall effects of time pressure, as well as individual differences, we identified the best-fitting values of free parameters for each participant's data separately, obtaining estimates of their posterior distributions using the differentially evolving Monte-Carlo Markov Chain (DEMCMC) sampling method[41,64]. In brief, this method simulates the likelihood of the observed data (i.e., choices and RTs) given a specific combination of parameters, and uses this likelihood to construct a Bayesian estimate of the posterior distribution of the likelihood of the parameters given the data. To maximise the data in estimating our model, we included missed responses by estimating the probability that simulations would fail to result in a choice prior to the time constraints imposed in the high

**Table 1 Scaling function, $f_{sc}(x)$, for each diffusion model parameter.**

| Parameter | $f_{sc}(x)$ | Min | Max |
|---|---|---|---|
| $w_{self}$ | $x \times 0.2 - 0.1$ | −0.1 | 0.1 |
| $w_{other}$ | $x \times 0.2 - 0.1$ | −0.1 | 0.1 |
| $w_{fairness}$ | $x \times 0.2 - 0.1$ | −0.1 | 0.1 |
| $b$ | $x \times 0.5$ | 0 | 0.5 |
| $d$ | $x \times 2.5$ | 0 | 2.5 |
| stbias | $x - 0.5$ | −0.5 | 0.5 |
| genbias | $x - 0.5$ | −0.5 | 0.5 |
| $\theta$ | $x$ | 0 | 1 |
| ndt | $x$ | 0 | 1 |

and low time pressure conditions. Missed trials constituted 4% and 0.3% of all trials in the high and low time pressure conditions, respectively.

For each subject fit, we ran $3 \times k$ chains in parallel[65], where $k$ is the number of free parameters, using uninformative priors. To preserve within-individual consistency in parameter values, we fitted data for both conditions simultaneously. In addition, to minimise the effects of parameters changing across individual fits of participants, we constrained possible parameter values as shown in Table 1, in accordance with values derived from previous model fits[26] and theoretical bounds.

Given these constraints, we employed a transformation in parameter sampling to ensure the prior distributions of the model parameters were truly uniform (noninformative) across the specified range in each condition. For a diffusion model with $k$ parameters ($k/2$ parameters for each of the 2 time pressure conditions), the DEMCMC sampler sampled $k$ MCMC model parameters comprising of $k/2$ individual predispositions ($M_{params}$) and $k/2$ effects of time pressure (change in parameters across conditions, $\delta_{params}$). The sum of these MCMC model parameters: $M_{params} + \delta_{params} \times$ Condition (effect-coded as high time pressure: 1; low time pressure: −1) is then transformed using an inverse probit transformation. The priors for the MCMC model parameters were specified as normal distributions with mean 0 and variance of 0.5, $N(0, 0.5)$ since inverse probit transformations of the sum of two normal distributions with mean 0 and variance of 0.5, $N(0, 0.5)$, yields a uniform distribution across values of [0, 1] for parameters in both conditions. These transformed values were then scaled by their respective functions, $f_{sc}(x)$, to the range of values as seen in Table 1 to derive the diffusion model parameter values.

To construct the estimated posterior distributions of each parameter, we sampled 500 iterations per chain after an initial burn-in period of 500 samples. For each iteration, the DEMCMC algorithm[41,64] proposes a new set of parameter estimates for each chain based on the scaled difference between the current parameter estimates of two other randomly sampled chains. The new parameter estimates are then evaluated by the Metropolis-Hastings algorithm for inclusion in the posterior distribution[64,66]. However, we included three additional features to our sampling method. Firstly, we implemented a probabilistic migration step, $\alpha = 0.1$, with every MCMC step in place of the differential evolution to improve chain-mixing and convergence towards the high probability density region of the posterior distribution of parameters. The migration step cycles the positions of a subset of chains ($N_{migrate}$ uniformly sampled from the total number of chains) such that the positions of chains $\{i, i + 1 \ldots j − 1, j\}$ were compared against $\{i + 1, i + 2, \ldots, j, i\}$ and evaluated based on the Metropolis-Hastings algorithm[64,66]. Secondly, we resampled likelihoods of the current parameter set of the chain, if proposals were rejected thrice consecutively, to avoid stuck chains and improve chain acceptance rates[41]. Lastly, we also reset MCMC chain positions halfway through the burn-in period (iteration 250) if they fell outside of the 95% confidence interval of the chain means. All chains were assessed to have converged to the posterior distribution with the Gelman-Rubin statistic, R-hat < 1.1[67], and the overall acceptance rate of proposals for each model ranged between [0.120 and 0.299].

To evaluate the likelihood of each parameter combination proposed by the DEMCMC given the observed choice data, we conducted 10,000 simulations of the candidate model given participant-specific proposal values and (for the gaze-informed ADDM) trial-specific fixation data. We then obtained the probability of observed choices (Yes/No) and RTs from these simulations using kernel density estimation techniques with a Gaussian kernel[68,69], obtaining smoothing bandwidths using Silverman's rule of thumb[70] multiplied by a factor of 0.5 to preserve the non-continuous nature of eye-movements, since saccades between AOIs could dramatically change the value evidence sampled, resulting in disjointed probability density distributions. Participant's parameters were estimated as the mean of the posterior distributions. To ensure adequate fits, we excluded the 3 participants with technical difficulties in eye-tracking and an additional seven participants who provided the same response more than 90% of the time (i.e., accepting or rejecting >90% of trials) leaving a final sample for model-fitting of $N = 50$.

To evaluate the fits of extracted model parameters in predicting the variability in behaviour within an individual participant, we conducted 10,000 simulations of each trial for each participant with the fitted diffusion model parameters to obtain the model-predicted average acceptance rate and reaction time of every choice trial.

To visualise and quantify these fits, we quantised both the choice data and reaction time data for each individual into 10 quantiles based on the model-predicted values of those measures per condition. We then computed each quantile mean of both observed and predicted values. Finally, we correlated the observed and predicted quantile means for both choice and reaction times for each of the time pressure conditions for each subject and performed a one-sample $t$ test on these correlation coefficients across subjects to evaluate the model's ability to capture variability in choice and reaction times within an individual.

**Model cross-validation.** To conduct cross-validation, we fitted versions of both the gaze-informed ADDM and a multi-attribute DDM to half of the data (odd trials) and tested model generated predictions on the other half of the data (even trials). To generate these predictions for testing, we conducted 10,000 simulations of each trial for each participant with the fitted diffusion model parameters to obtain the model-predicted average. We then evaluated whether the various models were able to capture the inter-individual variability in out-of-sample generosity under high time pressure, low time pressure and the observed relative change in generosity using correlations. To compare between models, we used Fisher's $t$ test of dependent correlations to identify models that were significantly more accurate in predicting the interindividual variability in the out-of-sample data.

**Software.** Continuous eye-tracking analyses were conducted using MATLAB (v2017b) and custom computational model fitting was conducted in Python (v3.6) based on kernel density estimation with SciPy (v1.4.1), parallelised with the native MATLAB Distributed Computing Server and the concurrent futures module in the Python Standard Library. All other statistical analyses were conducted in R 3.6.0. General linear mixed effects modelling was conducted using the lme4 package (v1.1-21) with degrees of freedom estimated using the Satterthwaite method. We conducted stepwise regression as implemented using the stepAIC function from the MASS package (v7.3-51.4). Effect size statistics of $R^2$ and partial $R^2$ were computed using the r2beta function from the r2glmm package (v0.1.2) using standardised general variances.

**Reporting summary.** Further information on research design is available in the Nature Research Reporting Summary linked to this article.

## Data availability

The experimental data that support the findings of this study are available in OSF with the identifier VF6A5 [https://doi.org/10.17605/OSF.IO/VF6A5]. Source data are provided with this paper. A reporting summary for this article is available as a Supplementary Information file.

## Code availability

All computer code generated for the computational models and analyses are available on OSF at with the identifier: VF6A5 [https://doi.org/10.17605/OSF.IO/VF6A5].

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

## Acknowledgements

This research was supported by an Insight Grant to C.A.H. from the Social Sciences and Humanities Research Council of Canada (SSHRC), grant no. 435-2016-1274. Computations were performed on the Niagara supercomputer at the SciNet HPC Consortium. SciNet is funded by: the Canada Foundation for Innovation; the Government of Ontario; Ontario Research Fund—Research Excellence; and the University of Toronto. We also thank members of the Toronto Decision Neuroscience Lab: Ian D. Roberts, Azadeh HajiHosseini, Daniel J. Wilson, Hause Lin, and Vignash Tharmaratnam for their helpful comments.

## Author contributions

Y.Y.T., Z.Y., W.A.C, and C.A.H. contributed to the preparation of the paper. C.A.H. and Z.Y. designed the eye-tracking study. Y.Y.T. and C.A.H. designed the online study. Y.Y.T. and Z.Y. analysed the data and Y.Y.T. developed the computational model. The scientific results and conclusions reflect the authors' opinions and not the views of SSHRC.

## Competing interests

The authors declare no competing interests.
