## [Peer Review File · Nature Communications]

Reviewers' comments:

Reviewer #1 (Remarks to the Author):

There is much to like about the paper. The authors develop a formal model for altruistic choice and they use eye-tracking data to test it.

The model fit of a complex model with 8 or 16 parameters was very good. The core behavioral finding of the studies is that there is no consistent influence of time pressure on pro-social choices. Instead, individuals who made more selfish choices under high time pressure (1.5 sec) became less selfish with more time to decide whereas individuals who made more pro-social choices under high time pressure became less pro-social. No such relation occurred with pro-sociality under low time pressure (10sec). They conclude that individuals' biases toward selfishness or pro-sociality only emerge under time pressure and are mitigated when people have more time to decide.

Although this work – and particularly the modelling - is clearly valuable, there are several problems in the argument of the authors and I therefore cannot recommend publication.

Major Issues

1) Embedding in theory and findings: There are hundreds of studies showing that persons without time pressure show consistent social preferences and that these influence their choices e.g. in dictator games. Conceptualizing these differences as biases that only appear under time pressure and are mitigated if sufficient time is in conflicts with most of the available evidence. It would require much stronger data for making such claims.

2) Internal validity: One alternative explanation for the behavioral results would be that the spontaneous responses are more noisy and people who behave more extremely show regression to the mean if more time is available.

3) Method: There is not independent measure of pro-sociality, which would be required to confirm the model and several of the authors' claims.

In replication study 1 the difference between high and low time pressure response time was 0.8 sec vs. 1.07 sec – this does not seem to be a major qualitative difference. I was wondering where this difference to the other studies came from. It seems that in the other studies people were reminded to delay their response if they answered before 2 sec. It remains unclear whether true deliberation or just delayed responding was induced.

4) Results: Attention dynamics look in general quite similar between high and low time pressure (Figure 3), except that in low time pressure there is a huge share of people looking somewhere else. If plotted as percentage against total fixations to AOI this would become better visible. Although differences in significance might appear, considering that people seemed to be forced to delay their choices anyway (see previous point) this might be merely due to increased noise in this early stage. Hence also these results seem not very convincing in support of the authors claims.

5) The author use their modelling results to derive conclusions concerning causality. This is not possible – plugging correlational data in a complex model does not allow for more valid conclusions concerning causality.

6) Modelling: the modelling is highly sophisticated and this could become the core contribution of this

work. Still, to do so it would require a comprehensive model comparison with flexibility correction, potentially also including the model by Chen & Krajbich (2018), that the authors merely aim to disqualify on methodological grounds in the online supplement. Cross-prediction test and many further standards for model comparisons would be required.

7) Referencing: The sources are in many cases relatively loosely selected and do not really support the points that are made. We for example know much more about who is pro-social spontaneously than implied in the introduction (i.e., persons with pro-social preferences drive the effect, e.g., Mischkowski & Glockner, 2016; Yamagishi et al., 2017). Just to take one example: In L406, the referencing is clearly misleading since most of the 7 cited studies do not show that time pressure robustly reveals differences in social preferences or at least not more so than measures without time pressure. Such a loose citation style should be avoided.

Reviewer #2 (Remarks to the Author):

The present manuscript investigates the effects of time pressure on attention and choices in mini-dictator games. A series of studies reveals that time pressure increases early attention to personal outcomes, and this early attention bias is correlated with future choices. However, there are individual differences in this effect, such that individuals with stronger gaze biases show a greater decrease in generosity.

Overall assessment

The paper is well-written and the research question is quite interesting. The paper makes a valuable contribution to the study of time pressure's effects on prosociality by incorporating eye-tracking data and presenting a new model that's supported by both empirical data and simulations. The data is also quite exciting, as it seems to contradict the predictions of the Social Heuristics Hypothesis, one of the main theories on how intuition influences cooperation. These results will be interesting to a broad, interdisciplinary audience! I have a few major comments, mainly dealing with how it is situated in the literature of dual process models of cooperation.

Comments

1. In the introduction (lines 43-53) the review of the literature may be a bit outdated.

a) Researchers are no longer interpreting self-paced response times as evidence of intuitive vs. reflective thinking (at least in the economic games literature) and the seemingly inconsistent results of prior studies can mainly be attributed to strength of preferences or feelings of conflict (e.g., Krajbich, Bartling, Hare, & Fehr, 2015).

b) The debate in studies using experimental designs (e.g., time pressure or intuition priming) seems to be about whether intuition increases cooperation (Rand, 2019), or has no significant effect on cooperative behavior (Bouwmeester et al., 2012).

2. It is unclear how the present studies fit in with the results of Chen and Krajbich (2019), who present evidence that time pressure can either increase or decrease prosociality; and propose also a modified DDM to account for the effects. The present studies do add a new component (namely, the eye tracking analyses) but seem to be closely replicating or extending on the results of Chen and Krajbich. The paper could be strengthened with a more explicit discussion of Chen and Krajbich, and clarification about how the new model differs from previous work. [Note that the pattern of results was also slightly different in the present study compared to Chen and Krajbich, who present results that are more in line with the “time pressure decisions are more extreme” account.]

3. The authors should do more to explicitly state how their findings present challenges for the Social Heuristics Hypothesis (SHH). In particular, there were two ways in which the present data don't fit with the prior research on the SHH:

A) The SHH explicitly hypothesizes that some people are intuitively cooperative and reflectively selfish, but no people are intuitively selfish and reflectively cooperative (Rand, Greene, & Nowak, 2012; Rand et al., 2014). The results clearly contradict this prediction: this is interesting and should get more attention.
B) The SHH further predicts that experience in economic games should eliminate the effects of time pressure (Rand et al., 2014). Once individuals have familiarity with the one-shot nature of the typical econ experiment, they should no longer be affected by the manipulation of time pressure. Again, the findings in the present experiment show a different pattern; participants complete many trials, and experience does not seem to attenuate time pressure effects.

4. Related to comment 3: There are two streams of studies on the effects of cognitive processes and prosociality. One approach (following the Rand et al., 2012 paper) focuses primarily on single shot experiments, focusing on between subject effects; in these studies, the time pressure manipulation is usually <10s for time pressure, >10s for forced delay. The other stream approaches the topic from a within-subject angle: participants complete many trials (sometimes hundreds each), and time pressure manipulations are on a much shorter scale (in this case < 1.5s vs. <10s). Decisions that are “delayed” in the present studies are comparable to the “time pressure” decisions in the Rand et al. paper. In the present studies, results are not in line with findings that follow the typical one-shot procedure. But the procedures differ in multiple substantive ways. How can we reconcile these two lines of research?

5. The authors contrast two ways that time pressure may affect individuals (lines 137-155). I found this difficult to follow. If I understand it correctly, there are two possible time pressure effects. A) time pressure decisions become more extreme (selfish people become more selfish; generous people become more generous); B) time pressure reveals individual differences that do not exist when there are no time constraints.

I am not sure that the authors can accurately differentiate between these two accounts with the given data. In the present studies, participants tended to be more selfish than generous (selfishness was consistently the dominant response; there were few participants who were generous more than 50% of the time). Thus, there would be relatively few participants who should actually become more generous under time pressure. It appears that both of these accounts would make similar predictions in the

present studies. (But I acknowledge that this may be a misinterpretation on my part.)

References

- Bouwmeester, S., Verkoeijen, P. P., Aczel, B., Barbosa, F., Bègue, L., Brañas-Garza, P., ... & Evans, A. M. (2017). Registered replication report: Rand, greene, and nowak (2012). *Perspectives on Psychological Science*, 12(3), 527-542.
- Chen, F., & Krajbich, I. (2018). Biased sequential sampling underlies the effects of time pressure and delay in social decision making. *Nature communications*, 9(1), 3557.
- Krajbich, I., Bartling, B., Hare, T., & Fehr, E. (2015). Rethinking fast and slow based on a critique of reaction-time reverse inference. *Nature communications*, 6, 7455.
- Rand, D. G. (2019). Intuition, Deliberation, and Cooperation: Further Meta-Analytic Evidence from 91 Experiments on Pure Cooperation. Available at SSRN 3390018.
- Rand, D. G., Greene, J. D., & Nowak, M. A. (2012). Spontaneous giving and calculated greed. *Nature*, 489(7416), 427.
- Rand, D. G., Peysakhovich, A., Kraft-Todd, G. T., Newman, G. E., Wurzbacher, O., Nowak, M. A., & Greene, J. D. (2014). Social heuristics shape intuitive cooperation. *Nature communications*, 5, 3677.

Reviewer #3 (Remarks to the Author):

This paper explores the mechanisms underlying cooperative decision-making when people are placed under time pressure. Specifically, the authors measure participants' eye gaze as a proxy for attention to self-oriented or other-oriented payoffs in order to quantify the influence of attention on subsequent decisions. They find that time pressure doesn't affect cooperative decision-making much overall, but that it accentuates people's individual biases towards selfishness or pro-sociality, and these biases are reflected in people's patterns of eye gaze.

The conclusions drawn from this work seem compelling and appropriately careful. As the authors hint at in the Discussion (406–408), it seems like these results suggest a role for both a categorically different type of cognition under time pressure and a fluid attention-based mechanism for gathering evidence in support of a decision. Specifically, the time pressure condition seems to induce more reliance on people's general biases, but then these biases also interact in interesting ways with the attentional dynamics of the participants.

While I'm compelled by these general conclusions, I must admit that some of the details of the results, particularly those from the DDM model, were difficult for me to follow. This likely reflects my relative unfamiliarity with these models, but I'm also left wondering how much a noisy 8-parameter model can teach us beyond what the qualitative results already show. I specifically worry about how the authors try to disentangle attention to outcomes and the weights put on those outcomes in their DDM model. Although I see how the model is meant to disambiguate these things, the two are presumably so

connected that it seems like one of these things may merely serve as a noisy proxy of the other. That is, because attention isn't manipulated in the study (it is endogenous), I suspect that people who, for example, have a high fitted w_{other} parameter but don't attend to others' outcomes much are actually just less prosocial than those who have a high w_{other} and do attend to others' outcome (cf. lines 377–378). In other words, the weights and attention terms are picking up on a common latent variable rather than measuring distinct features of some cognitive process. I'm not sure how much of a difference this makes for the authors' conclusions, but it does make me question how much the model can actually get at cognitive mechanism.

A more general worry I had with the model is the lack (as far as I can tell) of cross-validation or any other kind of out-of-sample metrics like WAIC. With 8 parameters, it's presumably not too hard to explain a lot of in-sample variance, but it's important to ensure that the model is not overfit to these data, especially when comparing the complex model to a simpler DDM. Again, my worry is that the authors are using this model to draw stronger inferences about cognitive process than are fully warranted. (This, by the way, is a worry I have with a lot of modeling work based on the DDM. Perhaps I'm just overly skeptical.)

Methodologically, it's been noted in time pressure studies that participants often fail to give a response in the allotted time. Dropping these trials can then induce a problematic selection bias when comparing remaining time pressure trials to time delay trials. As far as I can tell, this paper doesn't report any dropout metrics or what was done with trials in which participants' failed to respond in time. And I find it hard to believe that there were no such trials, given that responses needed to be given extremely quickly. It's important for the authors to report how these dropouts were handled, since this may also help explain why there was higher extremity in behavior under time pressure (e.g., maybe these were trials in which participants were most confident in their responses and could therefore act quickly).

A more minor point: the eye-gaze analyses seem to use ad hoc windows of time (see, e.g., 166–184). The authors might want to consider doing analyses with time as a continuous predictor. (A method like LOESS might be appropriate for tracking the dynamics of bias here.)

In short, the paper seems thorough and important, and I generally support publication, but I found some of the results and modeling opaque. It may be helpful for the authors to provide more detail on how the model supports the overall conclusions of the paper and verify that it is robust out-of-sample. The dropout rate of time-pressure trials is also important to look into.

Reviewers' comments:

Reviewer #1 (Remarks to the Author):

There is much to like about the paper. The authors develop a formal model for altruistic choice and they use eye-tracking data To further test it.

The model fit of a complex model with 8 or 16 parameters was very good. The core behavioral finding of the studies is that there is no consistent influence of time pressure on pro-social choices. Instead, individuals who made more selfish choices under high time pressure (1.5 sec) became less selfish with more time to decide whereas individuals who made more pro-social choices under high time pressure became less pro-social. No such relation occurred with pro-sociality under low time pressure (10sec). They conclude that individuals' biases toward selfishness or pro-sociality only emerge under time pressure and are mitigated when people have more time to decide.

Although this work – and particularly the modelling - is clearly valuable, there are several problems in the argument of the authors and I therefore cannot recommend publication.

We thank the reviewer for their encouraging remarks and extremely helpful comments on the paper, both here and below. We have taken these comments to heart in revising the framing, methods, results and discussion of the paper and believe the paper is much improved as a result.

Major Issues

1) Embedding in theory and findings: There are hundreds of studies showing that persons without time pressure show consistent social preferences and that these influence their choices e.g. in dictator games. Conceptualizing these differences as biases that only appear under time pressure and are mitigated if sufficient time is in conflicts with most of the available evidence. It would require much stronger data for making such claims.

We regret any confusion regarding our claims that biases appear only under time pressure. We did not intend to refute the extensive evidence in the literature showing that individuals show consistent social preferences that influence their choices. Indeed, our model explicitly assumes that those underlying preferences exist, both under time pressure and under free response conditions, and can drive attentional priorities in a way that manifest more strongly under time pressure (see for example our second prediction, which is that attentional biases reflect individual differences in concern for others). However, we also try to make the point that existing choice biases alone are not what drive *change* in choice under time pressure, but are mediated via strategic shifts in attention. It is these shifts in attention that represent a separable source of individual differences that appears to drive changes in choice behavior under time pressure. We thank the reviewer for their feedback and the opportunity to clarify this point and have amended the following sections to better communicate our intended argument.

Introduction (final paragraph): Here, we sought to emphasize and clarify that we believe individual differences in social preference exist and are an important factor driving our presumed effects. Specifically (changes highlighted, citations quoted in APA style for ease of reference):

“... Determining whether changes in gaze were driven by underlying priorities (which might represent a core individual difference present under both time pressure and free response conditions) requires some way to measure those priorities. However, if social preferences drive attention and attention drives the very choices used to infer preferences, this leads to a circularity of inference that makes causal analysis difficult. Thus, to identify underlying preferences independent of the effect of attention on choice, and to determine how those preferences might shape eye gaze, we developed a novel extension of attentional drift diffusion models (Krajbich, 2018; Krajbich, Armel, & Rangel, 2010; Smith & Krajbich, 2019) to simultaneously incorporate and account for real-time eye movements during choice. As expected, results from our computational model showed that individuals exhibit stable social preferences that predicted generosity across both conditions. Importantly, we found that the model also predicted both early attention biases as well as how these biases changed under time pressure. Further confirming our hypotheses, individual differences in attention interacted with time pressure and dispositional social preferences to predict changes in generosity, while potential markers of intuition-driven preferences did not. Finally, a follow-up study showed that forcing individuals to look at others’ outcomes rather than their own increased generosity, but only under time pressure, illustrating both the power and the limits of attention’s causal influence on choice. Thus, our model suggests that altruistic choice dynamics may result from dynamic attentional selection as opposed to the sequential activation of intuitive and controlled processes (Chen & Krajbich, 2018; Rand, Greene, & Nowak, 2012).”

Results: Processing time moderates individual differences in generosity. Here, we sought to clarify that we think gaze parameters might reflect not the only bias, but an additional bias that can influence choice.

“... Rather, our results suggest an additional source of individual variance that only emerged under time pressure and was mitigated when people had more time to decide. We hypothesized that these biases reflect changes in early attention’s influence on choice under time pressure and therefore sought to understand the mechanisms underlying these changes.”

Results: Absolute Model Fit. Here, we sought to clarify that our computational model largely confirms that there are stable individual differences that carry over from one condition to the other, and that these individual differences predict a large portion of the variability in generous responding.

“...Using logistic models, we also verified that model-extracted parameters quantifying dispositional social preferences (i.e. w_{self} , w_{other} , w_{fair}) were independently predictive of proportion generosity under high time pressure (w_{self} : $b = -41.120$, $SE = 3.837$, $z = -10.003$, $p < .001$, $R^2 = 0.276$; w_{other} : $b = 74.418$, $SE = 3.837$, $z = 19.393$, $p < .001$, $R^2 = 0.656$; w_{fair} : $b = 32.008$, $SE = 8.322$, $z = 3.846$, $p < .001$, $R^2 = 0.062$) and low time pressure (w_{self} : $b = -27.239$, $SE = 3.794$, $z = -7.179$, $p < .001$, $R^2 = 0.120$; w_{other} : $b = 60.077$, $SE = 3.507$, $z = 17.129$, $p < .001$, $R^2 = 0.536$; w_{fair} : $b = 29.469$, $SE = 7.921$, $z = 3.720$, $p < .001$, $R^2 = 0.044$).”

2) Internal validity: One alternative explanation for the behavioral results would be that the

spontaneous responses are more noisy and people who behave more extremely show regression to the mean if more time is available.

Like the reviewer, we acknowledge the potential for noise to alter choice behavior in significant ways. This is precisely the motivation in our choice to use drift-diffusion modelling to capture the independent influence of noise and preference in choice behavior. However, we think it unlikely that the proposed explanation (that people are more noisy under time delay rather than under time pressure) is the best account of our data, for several reasons. First and foremost, existing work on the effect of time on decision-making processes robustly show that the effects of noise are *attenuated* with more time, rather than enhanced (Bogacz, Brown, Moehlis, Holmes, & Cohen, 2006; Hawkins, Forstmann, Wagenmakers, Ratcliff, & Brown, 2015; Milosavljevic, Malmaud, Huth, Koch, & Rangel, 2010; Tajima, Drugowitsch, & Pouget, 2016).

Moreover, in our own data, we find consistent evidence that noise decreases with time. Specifically, fitted model parameters from our gaze-informed DDM reveal increases in the decision boundary ($M_{\text{delay}} = 0.396$, $M_{\text{pressure}} = 0.322$, $SE = 0.0157$, $t_{49} = 4.661$, $p < .001$, Cohen's $d = 0.66$) and decreases in boundary collapse rate ($M_{\text{delay}} = 0.203$, $M_{\text{pressure}} = 1.464$, $SE = 0.0603$, $t_{49} = -20.929$, $p < .001$, Cohen's $d = 2.96$) under time-delay compared to time-pressure. These increases in decision boundary have the effect of decreasing the effects of internal noise on the evidence signal during choice processes.

Additionally, increased noise during decision-making processes should result in choices that are less sensitive to the attributes of the choice. This is not what we observe in our data. Using logistic mixed-effects regression to predict whether individuals chose the proposal over the default amounts presented to them, we regressed their choice against their gains/losses in self-outcomes, gains/losses in other outcomes, time pressure as well as the two-way interactions between self-outcomes and time pressure, and between other-outcomes and time pressure. Consistent with the attenuation of attributes' impact on choice under time pressure, we find that the regression weights on self-outcomes and other-outcomes are **smaller under time pressure** ($\beta_{\text{self}} = 0.233$, $SE = 0.029$, $z = 8.024$, $p < .001$; $\beta_{\text{other}} = 0.023$, $SE = 0.033$, $z = 0.715$, $p = .47$) than under time-delay ($\beta_{\text{self}} = 0.302$, $SE = 0.029$, $z = 10.332$, $p < .001$; $\beta_{\text{other}} = 0.119$, $SE = 0.033$, $z = 3.619$, $p < .001$). These differences between the time pressure conditions were significant as shown by the two-way interactions between self-outcomes and time pressure ($\beta_{\text{self:pressure}} = -0.069$, $SE = 0.015$, $z = -4.775$, $p < .001$), and other-outcomes and time pressure ($\beta_{\text{other:pressure}} = -0.095$, $SE = 0.012$, $z = -7.861$, $p < .001$). We note that this analysis highlights an advantage of computational approaches: by including noise as an explicit feature of the model, and by examining the complex relationship between external evidence, choice, and reaction time, we are able to identify the extent to which the decreased coefficients in a regression model derive from a change in preferences vs. a change in sensitivity to noise (driven in part by a decrease in the threshold for choice).

Given that time pressure increases noise, and existing work suggests that noise would result in less consistent (i.e. more noisy) choices (Bogacz et al., 2006; Hawkins et al., 2015; Milosavljevic et al., 2010; Olschewski, Rieskamp, & Scheibehenne, 2018; Tajima et al., 2016), we think our pattern of results showing more extreme choice patterns under time pressure that attenuate with time are unlikely to be due to changes in noise in the decision process. Given that the paper is

already fairly long, we have currently opted not to include a discussion of this issue in the paper. We would, however, be happy to include a discussion of this, either in the main body of the paper or in a supplemental section, if the editor or reviewer think this would contribute substantially to the arguments we are making.

3) Method: (a) *There is not independent measure of pro-sociality, which would be required to confirm the model and several of the authors' claims.*

We share the reviewer's interest in linking our results to an independent measure of pro-sociality. While we agree that an independent measure of prosocial behavior would certainly add to the strength of our argument, the literature has shown that many independent measures of real-world helping are highly sensitive to context (Galizzi & Navarro-Martínez, 2018), making them less-than-ideal measures for the current purpose. Additionally, other potentially relevant dispositional measures such as psychopathy and empathy include measurement items unrelated to prosociality. Across multiple studies conducted in our lab, we have found little evidence that these related measures (empathy, psychopathy, real-world social behavior) predict generosity in these highly anonymized dictator games. We suspect this is true of other labs as well, since there is a comparatively sparse published literature on consistent relationships between anonymous dictator-game giving and real-world individual differences. And indeed, while we did include individual difference measures in our studies here, we failed to find a significant relationship between laboratory giving and self-report.

There is one individual difference measure that has become a popular measure of prosociality in the literature, and has shown some consistent prediction of dictator game behaviour in other studies: the Social Value Orientation scale (SVO: Murphy, Ackermann, & Handgraaf, 2011; Van Lange, 1999). However, we note that this measure utilizes game-theoretic approaches with items that comprise of discrete economic games that are nearly identical to the dictator games used in our studies, making additional measures of SVO essentially redundant as an independent measure. For this reason, we did not include this measure in our battery of individual difference questionnaires.

However, to assess the predictive accuracy of our model more generally, we have since conducted split-half cross validation by fitting our model to half of our data (odd trials) and testing the model-generated predictions of generosity in the other half (even trials). We show that our model achieves high out-of-sample predictive accuracy of both generosity and effects of time pressure on generosity within the dictator game. Specifically, the model was accurate at not only predicting individual differences in generosity under high time pressure (Pearson's $r = 0.911$, $t_{48} = 15.324$, $p < .001$) and low time pressure (Pearson's $r = 0.859$, $t_{48} = 11.630$, $p < .001$). It also accurately predicted the change in generosity between the conditions (Pearson's $r = 0.483$, $t_{48} = 3.818$, $p < .001$). We now report these analyses as part of the Model Comparison section of the results (see pg. 18-20).

Additionally, in a follow-up study conducted online, we found that generosity in these iterative binary choice dictator games was associated with giving in a one-shot dictator game where participants were given a total sum of money and asked to spontaneously generate how much they wished to allocate to their partner (Pearson's $r = 0.447$, $t_{198} = 7.029$, $p < .001$). Given that

these one-shot games are more similar to real-world giving opportunities, this suggests that binary-choice games tap into a similar underlying process. We have opted to explore this more extensively in Supplementary Note 3 since our paper is already fairly long.

(b) In replication study 1 the difference between high and low time pressure response time was 0.8 sec vs. 1.07 sec – this does not seem to be a major qualitative difference. I was wondering where this difference to the other studies came from. It seems that in the other studies people were reminded to delay their response if they answered before 2 sec. It remains unclear whether true deliberation or just delayed responding was induced.

This point identifies two issues: the difference between replication study 1 and the other studies, and the question of whether the free response condition actually induces greater deliberation. Regarding differences between replication study 1 and the others: we apologize for the lack of clarity in our description of the time-delay instructions for primary study and replication study 2. In the primary study and replication study 2, participants were instructed, “Try to think carefully about your choice, and choose in a way that reflects your preferences as accurately as possible”. Furthermore, when responses were faster than 2s, they were notified that they responded “fairly quickly” and reminded, “Take your time to make the best choice”. These instructions are consistent with prior manipulations, which have instructed participants to “think carefully about their decision before making it”, informed them that they “must wait at least 10 seconds before entering their decision or else they would not be allowed to participate” (Bouwmeester et al., 2017; Rand et al., 2012), and provided warnings to participants who responded too quickly or too slowly (Chen & Krajbich, 2018). Replication study 1 did not include quite such extensive measures, and so the effects on response time in this study were comparatively smaller. However, we emphasize that replication study 1 replicated the observed effects in the primary study and replication study 2, suggesting that effects observed in our primary study cannot be attributed simply to delayed responding. If anything, the fact that the same behavioural effect is observed across three different studies, using somewhat different time pressure manipulations, and resulting in somewhat different degrees of time pressure, speaks to the robustness of our findings. However, we thank the reviewer for their feedback and have amended the relevant methods section describing Study 1 and the replications to more clearly communicate the experimental manipulation (see pg. 29-30).

Regarding the second point about whether our manipulation resulted in increased deliberation: we suspect the reviewer is correct that some of the results we observe, particularly in the difference in response times, may be partially a function of a simple delay in responding. However, we think that several patterns observed in our data, particularly the eye-tracking measures, suggest that the free response condition resulted in at least some increased deliberation, rather than simply a delay in responding. First and foremost, if participants performed exactly the same computations in the time pressure and time free condition, but just introduced a delay in their responses, then we should not observe any of the systematic effects of time pressure on generosity that we did. We should have observed only a main effect on RT. Yet this was clearly not the case. Second, the eye tracking measures suggest that participants made a larger overall number of fixations to self and other information (Poisson mixed-effects regression on number of fixations per trial: $b_{low-high\ time\ pressure} = 1.081$, $SE = 0.0132$, $z = 82.16$, $p < .001$), and that the durations of those fixations were longer (linear mixed-effects regression on log-

transformed average duration of fixations per trial: $b_{low-high\ time\ pressure} = 0.317$, $SE = .00668$, $t_{158} = 47.51$, $p < .001$), when they had more time. This suggests that participants processed more choice information and/or processed the information more fully. Moreover, if people simply stopped deliberating at some point in the free time condition, we would expect a marked rise in fixations in non-AOI regions of the screen over time, and yet this is not what we observe. Gaze remains predominantly fixated on either self or other information until $\sim 7s$ where more than 95% of trials have terminated, suggesting extended deliberation of both self and other outcomes under time-delay.

Figure: Attention dynamics of altruistic choice across time. Millisecond-to-millisecond proportion of gaze directed to Self AOI, Other AOI or neither under low time pressure. The vertical lines represent points at which $> 50\%$ of trials (grey) and $> 95\%$ of trials (black) have terminated.

In order to keep the paper streamlined, since it is already fairly long, we have again opted not to include a full discussion of these additional analyses in the paper. However, we have included a reference to this in the paper (see pg. 29) and have added a fuller discussion of this issue in the supplementary materials (Supplementary Note 4).

4) Results: Attention dynamics look in general quite similar between high and low time pressure (Figure 3), except that in low time pressure there is a huge share of people looking somewhere else. If plotted as percentage against total fixations to AOI this would become better visible. Although differences in significance might appear, considering that people seemed to be forced to delay their choices anyway (see previous point) this might be merely due to increased noise in this early stage. Hence also these results seem not very convincing in support of the authors claims.

We acknowledge the reviewer's concern that forced-delay may result in increased variability in participants' initiation of the information acquisition process, resulting in "noisier" patterns of gaze biases. However, we think this is unlikely to explain our results. In particular, separate analyses of first fixation counts to one of the two AOIs (which simply reflect *what* information is acquired first, rather than exactly when) reveal similar patterns of results. Here, using fixation counts reduces the likelihood of measurement error that might be increased by noise from differences in the timing rather than nature of information search. In these analyses, we quantify

attention biases as the difference between proportion of first fixations on self-outcomes and proportion first fixations on other-outcomes. We replicate the finding reported in the paper that participants were biased towards self-information under time pressure ($b_0 = 1.395$, $SE = 0.332$, $z = 4.198$, $p < .001$) but not as strongly biased with time-delay ($b_0 = 0.612$, $SE = 0.331$, $z = 1.484$, $p = .065$). Additionally, the difference in first fixation biases between the time pressure and time delay condition is also significant ($b = 0.783$, $SE = 0.062$, $z = 12.702$, $p < .001$). All other results, including the effects of attention on generosity, are also similar if instead first-fixations alone are used. We have included a sentence in the Methods section to this effect (see pg. 13) as well as a supplementary note that details these analyses (Supplementary Note 1). We believe these analyses should allay any concern that these results are simply due to delays in processing or increases in noisy processing, rather than shifts in attentional priorities.

5) The author use their modelling results to derive conclusions concerning causality. This is not possible – plugging correlational data in a complex model does not allow for more valid conclusions concerning causality.

The reviewer's critique here directly addresses the crux of our thesis. We appreciate their feedback and have addressed this concern in several ways to improve our argument. First, we used the model itself to perform a stronger test of attention's causal effects. Specifically, we randomly permuted the order of fixations in our gaze data resulting in a data set where trial-level gaze durations and choice are equivalently correlated, but the within-trial temporal dynamics of this association is disrupted. If the order of fixations was *not* a causal influence on choice, then we would expect little difference in the predictive validity of a model that includes all fixations but in a jumbled order. If on the other hand, the dynamics of attention (and in particular the order of attention) matters in a causal way, as our hypothesis entails, then this model should not fit the data as well, particularly for the time pressure condition. We thus fit our gaze-informed ADDM to the permuted data. Using split-half cross-validation training on odd trials and validating testing against even trials, we find that the model that preserves the temporal order of fixations predicted out-of-sample data better than the model with permuted gaze data. Specifically, the model trained on the original data had better out-of-sample prediction accuracy for changes in generosity under time pressure (Pearson's $r = 0.483$, $t_{48} = 3.818$, $p < .001$) compared to the model trained on the scrambled data (Pearson's $r = 0.380$, $t_{48} = 2.850$, $p = .0064$; Fisher's t-test of dependent correlations = 1.65, one-tailed $p = .05$). Moreover, we obtained better overall choice prediction accuracy for generosity under high time pressure using the original model (Original: Pearson's $r = 0.911$, $t_{48} = 15.324$, $p < .001$; Scrambled: Pearson's $r = 0.879$, $t_{48} = 12.765$, $p < .001$; Fisher's $t = 2.75$, one-tailed $p < .001$). This was not true of predictions of generosity under low time pressure (Original: Pearson's $r = 0.859$, $t_{48} = 11.630$, $p < .001$; Scrambled: Pearson's $r = 0.855$, $t_{48} = 11.446$, $p < .001$; Fisher's $t = 0.28$ one-tailed $p = .39$).

We think these results strongly support the hypothesis of a causal influence of gaze on choice. While gaze is not experimentally manipulated within these studies, we specify a causal relationship between gaze and choice behavior within the model, such that gaze at each millisecond causally determines the evidence evaluated by the accumulator model. Additionally, the quality of model fit is evaluated only through the likelihood that the model generated predictions of choice behavior match the observed data. Given these specifications, and the results of the model comparison above, the specific millisecond-by-millisecond temporal

dynamics of attention clearly support a causal interpretation. Additionally, the extant literature on attention and choice supports some causal role of attention in the accumulation process in other domains of decision-making (for a review, see Orquin & Mueller Loose, 2013). Considering these previous findings, we believe that our results support a causal interpretation of the association between attention and its effects on choice.

Although we believe our data supports a causal account, we went one step further to address the reviewer's concern and conducted a follow-up study in which we experimentally manipulated attention during choice. In this study we find confirmatory evidence that attention drives generous choice, particularly under time pressure (interaction $b = -0.762$, $SE = 0.188$, $z = -4.061$, $p < .001$). Specifically, forcing people to look at other-outcomes compared to self-outcomes first predicted more generous behavior under high time pressure (simple $b = 0.811$, $SE = 0.143$, $z = 5.653$, $p < .001$) but not under low time pressure (simple $b = 0.049$, $SE = 0.121$, $z = 0.407$, $p = .68$).

We have included these new findings in a section "Forced attention to others' outcomes increases generosity under time pressure", copied below for convenience.

Results: Forced attention to others' outcomes increases generosity under time pressure

“ Our model suggests a causal account in which attention, driven in part by social preferences, directly influences the choices people make, especially under time pressure. However, the evidence we provided so far is largely correlational. Thus, to further test attention's causal influence on altruistic choice, we conducted a second, on-line study where we manipulated participant's attention to \$Self and \$Other. In this study, participants ($N = 200$) completed similar dictator games under high and low time pressure (see Methods for more details), using mouse clicks to reveal choice attributes (\$Self or \$Other) in a manner that approximated eye gaze in Study 1. Importantly, on some trials, they could choose which attribute to click on first. On other trials, they were forced to click on either \$Self or \$Other first. For parallelism with Study 1, we use the term “fixation” to describe when participants clicked on an attribute.

This design allowed us to test the causal importance of attention in a manner complementary to Study 1. If early attention derives its influence on choice only through its relationship with dispositional social preferences, first fixations should have little to no effect when they are exogenously controlled. In contrast, if attention has a direct causal influence on choice, then it should influence behavior even when determined by outside forces. To examine this possibility, we used logistic mixed-effects regression, predicting generous choice from first fixation, time pressure condition and their two-way interaction, separately for freely chosen and forced fixations. All analyses also controlled for the proportion of subsequent fixations on self-relative to other-outcomes and its two-way interaction with time pressure.

Replicating our finding from Study 1, participants were more likely to choose selfishly if they freely chose to look first at \$Self, specifically under time pressure (interaction: $b = -0.663$, $SE = 0.333$, $z = -1.995$, $p = .046$; simple effect of fixation under high time pressure: $b = 0.494$, $SE = 0.285$, $z = 1.736$, $p = .083$; simple effect of fixation under low time pressure: $b = -0.169$, $SE = 0.190$, $z = -0.894$, $p = .37$, Fig. 7a). Moreover, generosity in forced attention trials (which provides a measure of dispositional social preferences controlling for attention) predicted how selfishly-oriented participants' freely chosen first fixations were, especially under time pressure

(interaction: $b = 0.627$, $SE = 0.222$, $z = 2.828$, $p = .005$; simple effect of generosity under high time pressure: $b = -1.570$, $SE = 0.380$, $z = -4.134$, $p < .001$; simple effect of generosity under low time pressure: $b = -0.943$, $SE = 0.379$, $z = -2.492$, $p = .013$). These findings corroborate results from Study 1 suggesting that dispositional social preferences drive attentional priorities, particularly under time pressure, and that these attentional priorities correlate with choice.

Finally, we confirmed a causal effect of attention on choice independent of social preferences: generosity was strongly influenced by whether participants were forced to fixate on \$Self or \$Other first, but only if they were under time pressure, (interaction: $b = -0.762$, $SE = 0.188$, $z = -4.061$, $p < .001$; simple effect of fixation under high time pressure: $b = 0.811$, $SE = 0.143$, $z = 5.653$, $p < .001$; simple effect of fixation under low time pressure: $b = 0.049$, $SE = 0.121$, $z = 0.407$, $p = .68$, Fig. 7b). Thus, even when holding dispositional social preferences constant, directing participants' attention towards their own or others' outcomes made them more selfish or more generous, respectively.

Figure 7: First fixation predicts generosity under time pressure but not time delay for **a)** free attention trials, and **b)** forced attention trials. Central line in boxplots indicate estimated means and upper and lower bounds indicate one standard error above and below the mean. The whiskers indicate the 95% confidence interval for the estimated mean. * $p < .05$, ** $p < .01$, *** $p < .001$ ”

6) Modelling: the modelling is highly sophisticated and this could become the core contribution of this work. Still, to do so it would require a comprehensive model comparison with flexibility correction, potentially also including the model by Chen & Krajbich (2018), that the authors merely aim to disqualify on methodological grounds in the online supplement. Cross-prediction test and many further standards for model comparisons would be required.

We thank the reviewer for their encouraging remarks regarding our computational model and acknowledge the need to implement rigorous cross validation and model comparison with alternative models. We have taken their comments to heart and conducted extensive cross-validation and model comparison with other models including an implementation of the Chen &

Krajbich (2018) model. To circumvent known limitations of flexibility correction, we conducted model comparison using out-of-sample predictive accuracy. We have added these analyses as a “Model Comparison” section of the article (copied for convenience below). As you will see, all of these results support the original conclusions of our paper.

Results: Model Comparison

“ To assess the added explanatory and predictive value of accounting for dynamic attention in our gaze-informed ADDM, we first fit a version of the “biased DDM” as specified in Chen & Krajbich (Chen & Krajbich, 2018) with no information about eye gaze and no attentional discount parameter. Using split-half cross-validation, we compared the two models on their out-of-sample predictive accuracy, both for change as a function of time pressure, as well as mean levels of generosity in each condition. Notably, we found that the biased DDM failed to predict changes due to time pressure (Pearson’s $r = 0.219$, $t_{48} = 1.558$, $p = .126$), while the gaze-informed ADDM was significantly more accurate (Pearson’s $r = 0.483$, $t_{48} = 3.818$, $p < .001$; comparison to biased DDM Fisher’s $t = 2.16$, one-tailed $p = .02$, see Fig. 6). We observed similar differences when examining predictive accuracy for mean levels of generosity in each condition separately. While both models accurately predicted generosity in the high time pressure condition (Gaze-informed ADDM: Pearson’s $r = 0.911$, $t_{48} = 15.324$, $p < .001$; Biased DDM: Pearson’s $r = 0.783$, $t_{48} = 8.730$, $p < .001$) and the low time pressure condition (Gaze-informed ADDM: Pearson’s $r = 0.859$, $t_{48} = 11.630$, $p < .001$; Biased DDM: Pearson’s $r = 0.702$, $t_{48} = 6.825$, $p < .001$), the gaze-informed ADDM was significantly more accurate in both cases (High time pressure: Fisher’s $t = 3.58$, one-tailed $p < .001$; Low time pressure: Fisher’s $t = 3.44$, one-tailed $p < .001$). Thus, while social response biases, such as those implemented in the biased DDM, may partially drive choice behavior, they cannot fully account for the effects of time pressure on altruistic choice. Instead, accounting for the temporal dynamics of attention seems to be **necessary** to fully capture the effects of time pressure on altruistic choice.

Figure 6: Model comparison between the gaze-informed ADDM and the biased DDM(Chen & Krajbich, 2018) in their predictive accuracy of changes in generosity under time pressure. Each point represents a subject (N = 50), and points on the dashed line indicate perfect prediction of the observed data

We next sought to determine whether accounting for attention is sufficient to explain the effects of time pressure, or whether inclusion of a social response bias improves model fit. Given our strong hypotheses around the role of attention, we expected the exclusion of this social bias parameter to have minimal influence on predictive accuracy. As expected, a more parsimonious version of the gaze-informed model that excluded a social response bias was as accurate in predicting changes in generosity across the time pressure conditions as the full gaze-informed ADDM (nested model: Pearson's $r = 0.500$, $t_{48} = 3.998$, $p < .001$, Fisher's $t = -0.29$, two-tailed $p = .77$). It was also as accurate in predicting generosity under low time pressure (nested model: Pearson's $r = 0.848$, $t_{48} = 11.091$, $p < .001$, Fisher's $t = 0.74$, two-tailed $p = .46$). However, while the more parsimonious model accurately predicted generosity under high time pressure (nested model: Pearson's $r = 0.883$, $t_{48} = 13.014$, $p < .001$), the full gaze-informed model was slightly but significantly more accurate (Fisher's $t = 2.14$, two-tailed $p = .04$). These results suggest that, while social response biases may contribute to choice behavior under time pressure, a more parsimonious model accounting only for the temporal dynamics of attention is *sufficient* to capture the effects of time pressure on *change* in altruistic choice.

Additional model simulations further suggested that if attention drives early values, models that do not incorporate information about eye-gaze could erroneously attribute rapid selfish or generous responses to a response bias, a feature that may explain previous work attributing effects of time pressure to prepared response biases (see Supplementary Note 2). Altogether, these results strongly suggest that the temporal dynamics of attention control play a critical role in driving the effects of time pressure on altruistic choice.”

7) Referencing: The sources are in many cases relatively loosely selected and do not really support the points that are made. We for example know much more about who is pro-social spontaneously than implied in the introduction (i.e., persons with pro-social preferences drive the effect, e.g., Mischkowski & Glockner, 2016; Yamagishi et al., 2017). Just to take one example: In L406, the referencing is clearly misleading since most of the 7 cited studies do not show that time pressure robustly reveals differences in social preferences or at least not more so than measures without time pressure. Such a loose citation style should be avoided.

We apologize for the confusion as a result of our referencing style and thank the reviewer for their feedback on the scope of our literature review. We have revised the introduction and discussion to reflect a more contemporary and accurate interpretation of the literature. As mentioned before, citations here are given in full APA style for ease of reference. IN the paper, they accord with the style at Nature Communciations.

Introduction:

“ Unfortunately, work on this question has led to conflicting results and conclusions. Using response times as a proxy for automaticity and control has suggested both that prosociality may be rapid and instinctual (i.e. prosocial choices are faster than selfish choices) (Rand, 2017; Rand et al., 2012, 2014) and that it requires lengthy deliberation (i.e. prosocial choices are slower

than selfish choices) (Lohse, 2016; Lohse, Goeschl, & Diederich, 2014; Moore & Loewenstein, 2004; Piovesan & Wengström, 2009). Although recent work has called into question inferences that can be drawn from deliberation times (Evans, Dillon, & Rand, 2015; Krajbich, Bartling, Hare, & Fehr, 2015; Yamagishi et al., 2017), stronger causal manipulations that attempt to interfere with controlled processing using time pressure or instructions to respond instinctively have also led to conflicting conclusions, with people sometimes becoming more selfish (Krawczyk & Sylwestrzak, 2018; Shalvi, Eldar, & Bereby-Meyer, 2012) and sometimes becoming more generous (Bouwmeester et al., 2017; Chen & Krajbich, 2018; Rand et al., 2012; Rand, Newman, & Wurzbacher, 2015; Rand et al., 2014). More recently, some evidence has argued that some individuals have intuitively generous dispositions, while others are more intuitively selfish (Chen & Krajbich, 2018; Mischkowski & Glöckner, 2016; Yamagishi et al., 2017). Yet other work suggests that changes in choice behavior may not necessarily reflect differences in preferences at all, but rather differences in choice precision (Hutcherson, Bushong, & Rangel, 2015; Milosavljevic et al., 2010; Olschewski et al., 2018). Thus a crucial set of questions remains unanswered despite more than a decade of work: when choosing to act altruistically, does generosity or selfishness come first, and if so, why (Bouwmeester et al., 2017; Chen & Krajbich, 2018; Hutcherson et al., 2015; Mischkowski & Glöckner, 2016; Yamagishi et al., 2017)?”

Discussion:

“...Our findings help to make sense of the growing body of literature suggesting that time pressure neither consistently increases nor decreases generosity (Bouwmeester et al., 2017; Krawczyk & Sylwestrzak, 2018; Rand et al., 2012), but instead reveals individual differences in social preferences (Chen & Krajbich, 2018; Mischkowski & Glöckner, 2016). Moreover, while some of our findings hint at the possibility of fast and slow systems of processing (Chen & Krajbich, 2018; Kahneman & Egan, 2011; Moore & Loewenstein, 2004; Rand, 2016; Rand et al., 2012), they point firmly to the deep importance of attentional priorities in the face of processing constraints.”

Reviewer #2 (Remarks to the Author):

The present manuscript investigates the effects of time pressure on attention and choices in mini-dictator games. A series of studies reveals that time pressure increases early attention to personal outcomes, and this early attention bias is correlated with future choices. However, there are individual differences in this effect, such that individuals with stronger gaze biases show a greater decrease in generosity.

Overall assessment

The paper is well-written and the research question is quite interesting. The paper makes a valuable contribution to the study of time pressure's effects on prosociality by incorporating eye-tracking data and presenting a new model that's supported by both empirical data and simulations. The data is also quite exciting, as it seems to contradict the predictions of the Social Heuristics Hypothesis, one of the main theories on how intuition influences cooperation. These results will be interesting to a broad, interdisciplinary audience! I have a

few major comments, mainly dealing with how it is situated in the literature of dual process models of cooperation.

We thank the reviewer for their encouraging remarks and extremely helpful comments on the paper, both here and below. We have taken these comments to heart in revising the framing, methods, results and discussion of the paper.

Comments

1. In the introduction (lines 43-53) the review of the literature may be a bit outdated.

a) Researchers are no longer interpreting self-paced response times as evidence of intuitive vs. reflective thinking (at least in the economic games literature) and the seemingly inconsistent results of prior studies can mainly be attributed to strength of preferences or feelings of conflict (e.g., Krajbich, Bartling, Hare, & Fehr, 2015).

b) The debate in studies using experimental designs (e.g., time pressure or intuition priming) seems to be about whether intuition increases cooperation (Rand, 2019), or has no significant effect on cooperative behavior (Bouwmeester et al., 2012).

We apologize for the gaps in our literature review and thank the reviewer for their helpful suggestions. We have revised the introduction to reflect a more contemporary understanding of the literature regarding the use of response times. However, while we acknowledge that the cooperation literature has primarily focused on predictions of the social heuristic hypotheses in contrast to a null, other areas of prosocial behavior have found some evidence for self-interest being processed automatically and enhanced by experimental manipulations (Krawczyk & Sylwestrzak, 2018; Shalvi et al., 2012). We have revised our introduction accordingly. Note: All citations quoted here are given in APA format for ease of reference. In the paper, they accord to the numbered citation style used by Nature Communications.

Introduction:

“ Unfortunately, work on this question has led to conflicting results and conclusions. Using response times as a proxy for automaticity and control has suggested both that prosociality may be rapid and instinctual (i.e. prosocial choices are faster than selfish choices) (Rand, 2017; Rand et al., 2012, 2014) and that it requires lengthy deliberation (i.e. prosocial choices are slower than selfish choices) (Lohse, 2016; Lohse et al., 2014; Moore & Loewenstein, 2004; Piovesan & Wengström, 2009). Although recent work has called into question inferences that can be drawn from deliberation times (Evans et al., 2015; Krajbich et al., 2015; Yamagishi et al., 2017), stronger causal manipulations that attempt to interfere with controlled processing using time pressure or instructions to respond instinctively have also led to conflicting conclusions, with people sometimes becoming more selfish (Krawczyk & Sylwestrzak, 2018; Shalvi et al., 2012) and sometimes becoming more generous (Bouwmeester et al., 2017; Chen & Krajbich, 2018; Rand et al., 2012, 2015, 2014). More recently, some evidence has argued that some individuals have intuitively generous dispositions, while others are more intuitively selfish (Chen & Krajbich, 2018; Mischkowski & Glöckner, 2016; Yamagishi et al., 2017). Yet other work suggests that changes in choice behavior may not necessarily reflect differences in preferences at all, but rather differences in choice precision (Hutcherson et al., 2015; Milosavljevic et al., 2010;

Olschewski et al., 2018). Thus a crucial set of questions remains unanswered despite more than a decade of work: when choosing to act altruistically, does generosity or selfishness come first, and if so, why (Bouwmeester et al., 2017; Chen & Krajbich, 2018; Hutcherson et al., 2015; Mischkowski & Glöckner, 2016; Yamagishi et al., 2017)?”

2. It is unclear how the present studies fit in with the results of Chen and Krajbich (2019), who present evidence that time pressure can either increase or decrease prosociality; and propose also a modified DDM to account for the effects. The present studies do add a new component (namely, the eye tracking analyses) but seem to be closely replicating or extending on the results of Chen and Krajbich. The paper could be strengthened with a more explicit discussion of Chen and Krajbich, and clarification about how the new model differs from previous work. [Note that the pattern of results was also slightly different in the present study compared to Chen and Krajbich, who present results that are more in line with the “time pressure decisions are more extreme” account.]

Reviewer 1 also commented on this issue. We acknowledge that the distinct but related model presented by Chen & Krajbich (2018) invites comparison with our proposed model. We have thus added a more extensive comparison and discussion of the distinct features of both models and their implications in the “The Gaze-Informed Attentional Drift Diffusion Model” & “Model Comparison” section, copied below for convenience.

Results: The Gaze-Informed Attentional Drift Diffusion Model

“ Our model formalises the intuitive notion implicit in dual process models that value construction of options unfolds over time, and that cutting off the evidence accumulation process through time pressure can alter choices. It considers **three** possible mechanisms for these changes. **The first mechanism (as hypothesized by a Social Heuristics (Rand et al., 2012, 2014) or dual process account)** assumes that early evidence can differ from later evidence, and that it does so because automatic processing generates evidence more rapidly than controlled processing (Diederich & Trueblood, 2018). Thus, forcing people to choose quickly pre-empts late-emerging evidence, more cleanly revealing the contents of automatic or intuitive preference (Moore & Loewenstein, 2004; Rand et al., 2012). This predicts that changes in weight parameters should predict changes in generosity, assuming that under time pressure, intuitively selfish individuals initially weight self-interest more highly while intuitively prosocial individuals initially weight outcomes of others more highly. **The second mechanism (derived from our prioritized attention model)** also hypothesizes differences in early vs. later evidence, but posits that this results not from fixed activation of internal processes, but the dynamic and strategic allocation of attention. This predicts that changes in attention, rather than changes in weight parameters, should relate to changes in generosity. Finally, **a third mechanism (i.e., the biased DDM proposed by Chen and Krajbich (Chen & Krajbich, 2018))** suggests that time pressure exacerbates the effect of response-related starting biases, whose influence on the position of the evidence accumulation process declines as more evidence is gathered. Such starting biases could produce an automatic “default” response towards generosity or selfishness (Chen & Krajbich, 2018). This predicts that the estimated parameter values for starting biases, after accounting for weights and attention, should predict changes in generosity under time pressure.”

Results: Model Comparison

To assess the added explanatory and predictive value of accounting for dynamic attention in our gaze-informed ADDM, we first fit a version of the “biased DDM” as specified in Chen & Krajbich (Chen & Krajbich, 2018) with no information about eye gaze and no attentional discount parameter. Using split-half cross-validation, we compared the two models on their out-of-sample predictive accuracy, both for change as a function of time pressure, as well as mean levels of generosity in each condition. Notably, we found that the biased DDM failed to predict changes due to time pressure (Pearson’s $r = 0.219$, $t_{48} = 1.558$, $p = .126$), while the gaze-informed ADDM was significantly more accurate (Pearson’s $r = 0.483$, $t_{48} = 3.818$, $p < .001$; comparison to biased DDM Fisher’s $t = 2.16$, one-tailed $p = .02$, see Fig. 6). We observed similar differences when examining predictive accuracy for mean levels of generosity in each condition separately. While both models accurately predicted generosity in the high time pressure condition (Gaze-informed ADDM: Pearson’s $r = 0.911$, $t_{48} = 15.324$, $p < .001$; Biased DDM: Pearson’s $r = 0.783$, $t_{48} = 8.730$, $p < .001$) and the low time pressure condition (Gaze-informed ADDM: Pearson’s $r = 0.859$, $t_{48} = 11.630$, $p < .001$; Biased DDM: Pearson’s $r = 0.702$, $t_{48} = 6.825$, $p < .001$), the gaze-informed ADDM was significantly more accurate in both cases (High time pressure: Fisher’s $t = 3.58$, one-tailed $p < .001$; Low time pressure: Fisher’s $t = 3.44$, one-tailed $p < .001$). Thus, while social response biases, such as those implemented in the biased DDM, may partially drive choice behavior, they cannot fully account for the effects of time pressure on altruistic choice. Instead, accounting for the temporal dynamics of attention seems to be *necessary* to fully capture the effects of time pressure on altruistic choice.

Figure 6: Model comparison between the gaze-informed ADDM and the biased DDM(Chen & Krajbich, 2018) in their predictive accuracy of changes in generosity under time pressure. Each point represents a subject (N = 50), and points on the dashed line indicate perfect prediction of the observed data

We next sought to determine whether accounting for attention is sufficient to explain the effects of time pressure, or whether inclusion of a social response bias improves model fit. Given our strong hypotheses around the role of attention, we expected the exclusion of this social bias parameter to have minimal influence on predictive accuracy. As expected, a more parsimonious version of the gaze-informed model that excluded a social response bias was as accurate in predicting changes in generosity across the time pressure conditions as the full gaze-informed ADDM (nested model: Pearson's $r = 0.500$, $t_{48} = 3.998$, $p < .001$, Fisher's $t = -0.29$, two-tailed $p = .77$). It was also as accurate in predicting generosity under low time pressure (nested model: Pearson's $r = 0.848$, $t_{48} = 11.091$, $p < .001$, Fisher's $t = 0.74$, two-tailed $p = .46$). However, while the more parsimonious model accurately predicted generosity under high time pressure (nested model: Pearson's $r = 0.883$, $t_{48} = 13.014$, $p < .001$), the full gaze-informed model was slightly but significantly more accurate (Fisher's $t = 2.14$, two-tailed $p = .04$). These results suggest that, while social response biases may contribute to choice behavior under time pressure, a more parsimonious model accounting only for the temporal dynamics of attention is *sufficient* to capture the effects of time pressure on *change* in altruistic choice.

Additional model simulations further suggested that if attention drives early values, models that do not incorporate information about eye-gaze could erroneously attribute rapid selfish or generous responses to a response bias, a feature that may explain previous work attributing effects of time pressure to prepared response biases (see Supplementary Note 2). Altogether, these results strongly suggest that the temporal dynamics of attention control play a critical role in driving the effects of time pressure on altruistic choice.”

3. The authors should do more to explicitly state how their findings present challenges for the Social Heuristics Hypothesis (SHH). In particular, there were two ways in which the present data don't fit with the prior research on the SHH:

A) The SHH explicitly hypothesizes that some people are intuitively cooperative and reflectively selfish, but no people are intuitively selfish and reflectively cooperative (Rand, Greene, & Nowak, 2012; Rand et al., 2014). The results clearly contradict this prediction: this is interesting and should get more attention.

B) The SHH further predicts that experience in economic games should eliminate the effects of time pressure (Rand et al., 2014). Once individuals have familiarity with the one-shot nature of the typical econ experiment, they should no longer be affected by the manipulation of time pressure. Again, the findings in the present experiment show a different pattern; participants complete many trials, and experience does not seem to attenuate time pressure effects.

We thank the reviewer for their insightful comment and acknowledge the implications our results have for the social heuristic hypotheses. We have revised the discussion section to include a more explicit discussion of the SHH and its predictions.

Discussion:

“...Yet these results appear to conflict with the idea that individuals often default to pro-social responses (Social Heuristic Hypothesis)(Rand et al., 2012; Yamagishi et al., 2017; Zaki & Mitchell, 2013). What might reconcile such findings? Two possibilities are immediately apparent.

First, much of the research demonstrating intuitive biases tends to use single-shot games in which participants self-generate proposed monetary splits, whereas our study used repeated presentation of specific, experimenter-determined payouts. Perhaps response modality, or the repeated exposure to games of this sort (Rand et al., 2014) results in different decision processes. However, in supplementary analyses of Study 2, we incorporated a series of one-shot games into our measure of prosociality and found that generosity in one-shot continuous vs. repeated binary choices were strongly correlated, and that this correlation was stronger when matched for conditions of time-pressure or time-delay (see Supplementary Note 3). These results suggest that one-shot and binary-choice games tap into a common set of processes.

Instead, we think a second explanation is more likely. Much of the literature demonstrating default pro-social biases has emerged during the study of strategic cooperative behaviour, such as the Ultimatum or Public Goods Game, rather than purely altruistic behaviour (Evans et al., 2015; Rand et al., 2012, 2014; Yamagishi et al., 2017). We suspect that when one's own outcomes are more clearly predicated on the behaviour or outcomes of others (as in the Ultimatum Game), attention shifts more decisively toward social information, yielding pro-social biases under time pressure. Future work using eye tracking could test such a hypothesis by comparing the pattern of eye gaze during Dictator and Ultimatum game proposals."

4. Related to comment 3: There are two streams of studies on the effects of cognitive processes and prosociality. One approach (following the Rand et al., 2012 paper) focuses primarily on single shot experiments, focusing on between subject effects; in these studies, the time pressure manipulation is usually <10s for time pressure, >10s for forced delay. The other stream approaches the topic from a within-subject angle: participants complete many trials (sometimes hundreds each), and time pressure manipulations are on a much shorter scale (in this case < 1.5s vs. <10s). Decisions that are "delayed" in the present studies are comparable to the "time pressure" decisions in the Rand et al. paper. In the present studies, results are not in line with findings that follow the typical one-shot procedure. But the procedures differ in multiple substantive ways. How can we reconcile these two lines of research?

We thank the reviewer again for their insightful comment and acknowledge the differences in experimental context between these divergent studies. Most crucially, the single-shot experiments often require participants to spontaneously generate a distribution of resources for themselves and their partners. However, the repeated-measures studies tend to require participants to iteratively choose between predefined alternatives. We speculate that the generative process in one-shot games may involve multiple iterations of generating candidate distributions (similar to the sort we use here), evaluating these candidates, and then making a choice. Thus, the processing required to make these choices possibly comprise of multiple sequential choices and the time pressure manipulations are appropriately less intensive to capture the task demand.

To empirically bridge this gap, we conducted a follow-up study where participants first made these one-shot dictator games where they generate a distribution on a continuous scale under time pressure or time-delay. These participants then made a series of binary-choices between predetermined distributions under time pressure and time delay. Importantly, we found that behaviors in these games tap similarly into underlying social preferences and decision-making

processes. We have now referred to this in the discussion of the paper (see pg. 27) and added a supplementary note that discusses this concern in further detail (Supplementary Note 3).

Thus, we believe this provides some evidence that the differences in results and interpretations of time pressure effects on choice are unlikely to be due to differences in measurement. Instead, as we outline in our discussion, we believe these disparities to be largely due to differences in the context of strategic cooperation and pure altruism.

5. The authors contrast two ways that time pressure may affect individuals (lines 137-155). I found this difficult to follow. If I understand it correctly, there are two possible time pressure effects. A) time pressure decisions become more extreme (selfish people become more selfish; generous people become more generous); B) time pressure reveals individual differences that do not exist when there are no time constraints.

I am not sure that the authors can accurately differentiate between these two accounts with the given data. In the present studies, participants tended to be more selfish than generous (selfishness was consistently the dominant response; there were few participants who were generous more than 50% of the time). Thus, there would be relatively few participants who should actually become more generous under time pressure. It appears that both of these accounts would make similar predictions in the present studies. (But I acknowledge that this may be a misinterpretation on my part.)

We thank the reviewer for their helpful comment and apologize for any confusion. We have since revised the relevant section in the results, shown below, to more clearly illustrate our argument. The reviewer correctly identifies the two possible individual differences in the effects of attention considered in our paper. We acknowledge that the robustness of our claims may not be appropriately supported in our main study due to an overall selfish bias in responding within our sample. To this end, we've conducted additional analyses pooling participants across the primary and supplementary studies (N = 174).

First, we divided the data into subsets of selfish and generous subjects based on their percentage generosity under high time pressure (selfish: < 45%; generous > 55%). We then conducted t-tests on their change in generosity from high to low time pressure. Here we find that selfish individuals became less selfish with time (M = 0.0502, SE = 0.0094, $t_{75} = -5.361$, $p < .001$) and generous individuals became less generous (M = -0.0295, SE = 0.0084, $t_{48} = -3.499$, $p = .0011$). This is consistent with our hypothesis that some individual differences in behavior under time pressure become attenuated with time.

Next, we divided the full dataset into subsets of selfish and generous individuals based on their percentage generosity under low time pressure (selfish: < 45%; generous > 55%). We then conducted t-tests on their change in generosity from low to high time pressure. Here we find that both selfish and generous individuals become more selfish under time pressure (selfish: M = 0.0272, SE = 0.0107, $t_{73} = -2.539$, $p = .013$; generous: M = 0.0376, SE = 0.0087, $t_{48} = -4.389$, $p < .001$), and no evidence that individual differences become exacerbated under time pressure.

Together with the correlational results reported in the main manuscript, we believe these results provide strong evidence to suggest that the individual differences in the change in generosity

under time pressure derives from a unique source of variance that emerges under high time pressure and attenuates with time. This effect is separable from choice behavior under low time pressure and likely driven by changes in attention as we show later in the paper. We have opted not to include these additional analyses due to length considerations. However, if the reviewer and the editor believe it adds to the strength of our argument, we would gladly include it in the main body of the paper or a supplementary note.

References

- Bouwmeester, S., Verkoeijen, P. P., Aczel, B., Barbosa, F., Bègue, L., Brañas-Garza, P., ... & Evans, A. M. (2017). Registered replication report: Rand, Greene, and Nowak (2012). *Perspectives on Psychological Science*, 12(3), 527-542.**
- Chen, F., & Krajbich, I. (2018). Biased sequential sampling underlies the effects of time pressure and delay in social decision making. *Nature communications*, 9(1), 3557.**
- Krajbich, I., Bartling, B., Hare, T., & Fehr, E. (2015). Rethinking fast and slow based on a critique of reaction-time reverse inference. *Nature communications*, 6, 7455.**
- Rand, D. G. (2019). *Intuition, Deliberation, and Cooperation: Further Meta-Analytic Evidence from 91 Experiments on Pure Cooperation*. Available at SSRN 3390018.**
- Rand, D. G., Greene, J. D., & Nowak, M. A. (2012). Spontaneous giving and calculated greed. *Nature*, 489(7416), 427.**
- Rand, D. G., Peysakhovich, A., Kraft-Todd, G. T., Newman, G. E., Wurzbacher, O., Nowak, M. A., & Greene, J. D. (2014). Social heuristics shape intuitive cooperation. *Nature communications*, 5, 3677.**

We thank the reviewers for these helpful references and have incorporated them into our literature review.

Reviewer #3 (Remarks to the Author):

This paper explores the mechanisms underlying cooperative decision-making when people are placed under time pressure. Specifically, the authors measure participants' eye gaze as a proxy for attention to self-oriented or other-oriented payoffs in order to quantify the influence of attention on subsequent decisions. They find that time pressure doesn't affect cooperative decision-making much overall, but that it accentuates people's individual biases towards selfishness or pro-sociality, and these biases are reflected in people's patterns of eye gaze.

The conclusions drawn from this work seem compelling and appropriately careful. As the authors hint at in the Discussion (406–408), it seems like these results suggest a role for both a categorically different type of cognition under time pressure and a fluid attention-based mechanism for gathering evidence in support of a decision. Specifically, the time pressure condition seems to induce more reliance on people's general biases, but then these biases also interact in interesting ways with the attentional dynamics of the participants.

We thank the reviewer for their encouraging remarks and extremely helpful comments on the paper, both here and below. We have taken these comments to heart in revising the framing, methods, results and discussion of the paper.

(1) While I'm compelled by these general conclusions, I must admit that some of the details of the results, particularly those from the DDM model, were difficult for me to follow. This likely reflects my relative unfamiliarity with these models, but I'm also left wondering how much a noisy 8-parameter model can teach us beyond what the qualitative results already show. I specifically worry about how the authors try to disentangle attention to outcomes and the weights put on those outcomes in their DDM model. Although I see how the model is meant to disambiguate these things, the two are presumably so connected that it seems like one of these things may merely serve as a noisy proxy of the other. That is, because attention isn't manipulated in the study (it is endogenous), I suspect that people who, for example, have a high fitted w_{other} parameter but don't attend to others' outcomes much are actually just less prosocial than those who have a high w_{other} and do attend to others' outcome (cf. lines 377–378). In other words, the weights and attention terms are picking up on a common latent variable rather than measuring distinct features of some cognitive process. I'm not sure how much of a difference this makes for the authors' conclusions, but it does make me question how much the model can actually get at cognitive mechanism.

We thank the reviewer here for their encouraging remarks and agree that it is often difficult to fully disentangle the variance associated with attention and evaluative weights. We also acknowledge that this may be a limitation of our design as we did not manipulate attention. Additionally, we do not exclude the possibility that evaluation and attention are picking up on a common latent variable.

Precisely, we believe that social preferences more generally drive these manifestations. Under low time pressure, social preferences manifest primarily in the evaluative weights due to ample opportunity for information gathering and few constraints on attentional search. However, under high time pressure, processing constraints drive the emergence of attentional search biases that prioritize preferred information to compensate for the shortened decision duration. To show that attention and evaluation have distinct patterns of influence over choice behavior, we have now conducted (1) extensive model comparison and (2) a second study where we manipulated attention.

In summary, our model comparisons (see Model Comparison section on pg. 18-20) strongly suggest that some variability in choice is highly specific to moment-to-moment attention deployment that changes under time pressure. Specifically, we find that a model not accounting for attention fails to predict changes in generous choice under time pressure, showing that evaluative weights on their own in insufficient to capture the all the variability in choice behavior.

Our follow-up study manipulating attention, also supports the distinct role of attention independent of dispositional social preferences in the choice process. In this study we find confirmatory evidence that attention drives generous choice, specifically under time pressure. Notably, forcing participants to look at self-outcomes first predicted less generous behavior under time pressure but not under time. Here, participant's attention is exogenously directed and thus, independent of their dispositional social preferences. These effects strongly support a distinct causal mechanism for attention's effect on choice processes independent of evaluative

weights. We have included these new findings in a section at the end of the results section “Forced attention to others’ outcomes increases generosity under time pressure”, copied below for convenience. Note: All citations quoted here are given in APA format for ease of reference. In the paper, they accord to the numbered citation style used by Nature Communications.

Results: *Forced attention to others’ outcomes increases generosity under time pressure*

“ Our model suggests a causal account in which attention, driven in part by social preferences, directly influences the choices people make, especially under time pressure. However, the evidence we provided so far is largely correlational. Thus, to further test attention’s causal influence on altruistic choice, we conducted a second, on-line study where we manipulated participant’s attention to \$Self and \$Other. In this study, participants (N = 200) completed similar dictator games under high and low time pressure (see Methods for more details), using mouse clicks to reveal choice attributes (\$Self or \$Other) in a manner that approximated eye gaze in Study 1. Importantly, on some trials, they could choose which attribute to click on first. On other trials, they were forced to click on either \$Self or \$Other first. For parallelism with Study 1, we use the term “fixation” to describe when participants clicked on an attribute.

This design allowed us to test the causal importance of attention in a manner complementary to Study 1. If early attention derives its influence on choice only through its relationship with dispositional social preferences, first fixations should have little to no effect when they are exogenously controlled. In contrast, if attention has a direct causal influence on choice, then it should influence behavior even when determined by outside forces. To examine this possibility, we used logistic mixed-effects regression, predicting generous choice from first fixation, time pressure condition and their two-way interaction, separately for freely chosen and forced fixations. All analyses also controlled for the proportion of subsequent fixations on self-relative to other-outcomes and its two-way interaction with time pressure.

Replicating our finding from Study 1, participants were more likely to choose selfishly if they freely chose to look first at \$Self, specifically under time pressure (interaction: $b = -0.663$, $SE = 0.333$, $z = -1.995$, $p = .046$; simple effect of fixation under high time pressure: $b = 0.494$, $SE = 0.285$, $z = 1.736$, $p = .083$; simple effect of fixation under low time pressure: $b = -0.169$, $SE = 0.190$, $z = -0.894$, $p = .37$, Fig. 7a). Moreover, generosity in forced attention trials (which provides a measure of dispositional social preferences controlling for attention) predicted how selfishly-oriented participants’ freely chosen first fixations were, especially under time pressure (interaction: $b = 0.627$, $SE = 0.222$, $z = 2.828$, $p = .005$; simple effect of generosity under high time pressure: $b = -1.570$, $SE = 0.380$, $z = -4.134$, $p < .001$; simple effect of generosity under low time pressure: $b = -0.943$, $SE = 0.379$, $z = -2.492$, $p = .013$). These findings corroborate results from Study 1 suggesting that disposition social preferences drive attentional priorities, particularly under time pressure, and that these attentional priorities correlate with choice.

Finally, we confirmed a causal effect of attention on choice independent of social preferences: generosity was strongly influenced by whether participants were forced to fixate on \$Self or \$Other first, but only if they were under time pressure, (interaction: $b = -0.762$, $SE = 0.188$, $z = -4.061$, $p < .001$; simple effect of fixation under high time pressure: $b = 0.811$, $SE = 0.143$, $z = 5.653$, $p < .001$; simple effect of fixation under low time pressure: $b = 0.049$, $SE = 0.121$, $z = 0.407$, $p = .68$, Fig. 7b). Thus, even when holding dispositional social preferences constant, directing participants’ attention towards their own or others’ outcomes made them

more selfish or more generous, respectively.

Figure 7: First fixation predicts generosity under time pressure but not time delay for **a)** free attention trials, and **b)** forced attention trials. Central line in boxplots indicate estimated means and upper and lower bounds indicate one standard error above and below the mean. The whiskers indicate the 95% confidence interval for the estimated mean. * $p < .05$, ** $p < .01$, *** $p < .001$ ”

(2) A more general worry I had with the model is the lack (as far as I can tell) of cross-validation or any other kind of out-of-sample metrics like WAIC. With 8 parameters, it’s presumably not too hard to explain a lot of in-sample variance, but it’s important to ensure that the model is not overfit to these data, especially when comparing the complex model to a simpler DDM. Again, my worry is that the authors are using this model to draw stronger inferences about cognitive process than are fully warranted. (This, by the way, is a worry I have with a lot of modeling work based on the DDM. Perhaps I’m just overly skeptical.)

We thank the reviewer for their helpful feedback and acknowledge the need for cross-validation in our manuscript. We have now conducted split-half cross-validation and reported the out-of-sample prediction accuracy in addition to various model comparisons in the “Model Comparison” section of the results as mentioned above.

Results: Model Comparison

“ To assess the added explanatory and predictive value of accounting for dynamic attention in our gaze-informed ADDM, we first fit a version of the “biased DDM” as specified in Chen & Krajbich (Chen & Krajbich, 2018) with no information about eye gaze and no attentional discount parameter. Using split-half cross-validation, we compared the two models on their out-of-sample predictive accuracy, both for change as a function of time pressure, as well as mean levels of generosity in each condition. Notably, we found that the biased DDM failed to predict changes due to time pressure (Pearson’s $r = 0.219$, $t_{48} = 1.558$, $p = .126$), while the gaze-informed ADDM was significantly more accurate (Pearson’s $r = 0.483$, $t_{48} = 3.818$, $p < .001$; comparison to biased DDM Fisher’s $t = 2.16$, one-tailed $p = .02$, see Fig. 6). We observed similar

differences when examining predictive accuracy for mean levels of generosity in each condition separately. While both models accurately predicted generosity in the high time pressure condition (Gaze-informed ADDM: Pearson's $r = 0.911$, $t_{48} = 15.324$, $p < .001$; Biased DDM: Pearson's $r = 0.783$, $t_{48} = 8.730$, $p < .001$) and the low time pressure condition (Gaze-informed ADDM: Pearson's $r = 0.859$, $t_{48} = 11.630$, $p < .001$; Biased DDM: Pearson's $r = 0.702$, $t_{48} = 6.825$, $p < .001$), the gaze-informed ADDM was significantly more accurate in both cases (High time pressure: Fisher's $t = 3.58$, one-tailed $p < .001$; Low time pressure: Fisher's $t = 3.44$, one-tailed $p < .001$). Thus, while social response biases, such as those implemented in the biased DDM, may partially drive choice behavior, they cannot fully account for the effects of time pressure on altruistic choice. Instead, accounting for the temporal dynamics of attention seems to be **necessary** to fully capture the effects of time pressure on altruistic choice.

Figure 6: Model comparison between the gaze-informed ADDM and the biased DDM(Chen & Krajbich, 2018) in their predictive accuracy of changes in generosity under time pressure. Each point represents a subject ($N = 50$), and points on the dashed line indicate perfect prediction of the observed data

We next sought to determine whether accounting for attention is sufficient to explain the effects of time pressure, or whether inclusion of a social response bias improves model fit. Given our strong hypotheses around the role of attention, we expected the exclusion of this social bias parameter to have minimal influence on predictive accuracy. As expected, a more parsimonious version of the gaze-informed model that excluded a social response bias was as accurate in predicting changes in generosity across the time pressure conditions as the full gaze-informed ADDM (nested model: Pearson's $r = 0.500$, $t_{48} = 3.998$, $p < .001$, Fisher's $t = -0.29$, two-tailed $p = .77$). It was also as accurate in predicting generosity under low time pressure (nested model: Pearson's $r = 0.848$, $t_{48} = 11.091$, $p < .001$, Fisher's $t = 0.74$, two-tailed $p = .46$). However,

while the more parsimonious model accurately predicted generosity under high time pressure (nested model: Pearson's $r = 0.883$, $t_{48} = 13.014$, $p < .001$), the full gaze-informed model was slightly but significantly more accurate (Fisher's $t = 2.14$, two-tailed $p = .04$). These results suggest that, while social response biases may contribute to choice behavior under time pressure, a more parsimonious model accounting only for the temporal dynamics of attention is *sufficient* to capture the effects of time pressure on *change* in altruistic choice.

Additional model simulations further suggested that if attention drives early values, models that do not incorporate information about eye-gaze could erroneously attribute rapid selfish or generous responses to a response bias, a feature that may explain previous work attributing effects of time pressure to prepared response biases (see Supplementary Note 2). Altogether, these results strongly suggest that the temporal dynamics of attention control play a critical role in driving the effects of time pressure on altruistic choice.”

(3) Methodologically, it's been noted in time pressure studies that participants often fail to give a response in the allotted time. Dropping these trials can then induce a problematic selection bias when comparing remaining time pressure trials to time delay trials. As far as I can tell, this paper doesn't report any dropout metrics or what was done with trials in which participants' failed to respond in time. And I find it hard to believe that there were no such trials, given that responses needed to be given extremely quickly. It's important for the authors to report how these dropouts were handled, since this may also help explain why there was higher extremity in behavior under time pressure (e.g., maybe these were trials in which participants were most confident in their responses and could therefore act quickly).

We agree that considering non-responses is important. In fact, our model explicitly incorporates non-responses into the fitting process (which we now highlight on pg. 38 of the revised manuscript). Using simulations, we were able to calculate the probability that a particular parameter combination would result in a missed trial and used it in further maximum likelihood calculations to fit the models, thereby using all the data collected to estimate the model parameters. However, we acknowledge that we need to do more to determine how much of an effect non-responses might have on our data. To address the reviewer's concern, we have reported the rates of missed responses in the Methods section of the paper. Across all experiments, we find $< 5\%$ of missed responses in both high and low time pressure condition. We thus think it unlikely that missed responses significantly influenced conclusions of the paper.

Model Estimation:

“To maximize the data in estimating our model, we included missed responses by estimating the probability that simulations would fail to result in a choice prior to the time constraints imposed in the high and low time pressure conditions. Missed trials constituted 4% and 0.3% of all trials in the high and low time pressure conditions respectively.”

Methods: Generous Choice

“Missed response trials (mean percentage of trials: high time pressure: $M_{\text{study1}} = 4.67\%$, $M_{\text{REP1}} = 3.03\%$, $M_{\text{REP2}} = 2.29\%$, $M_{\text{study2}} = 1.03\%$; low time pressure: $M_{\text{study1}} = 0.33\%$, $M_{\text{REP1}} = 0.03\%$, $M_{\text{REP2}} = 0.39\%$, $M_{\text{study2}} = 0.03\%$) were excluded from choice analyses.”

(4) A more minor point: the eye-gaze analyses seem to use ad hoc windows of time (see, e.g.,

166–184). The authors might want to consider doing analyses with time as a continuous predictor. (A method like LOESS might be appropriate for tracking the dynamics of bias here.)

We agree with the reviewer that identifying the appropriate time-windows of interest is not trivial. However, we point out that the window of time used for the eye-gaze analyses was not ad hoc, but identified as the specific window of time during the early phases of choice in which gaze patterns differed between the time pressure conditions. These windows were identified using univariate logistic mixed effects regressions at each time point to identify time points of significant difference, and then cluster-corrected for multiple comparisons to identify the relevant time windows in which gaze was biased. We believe this method to be optimal to capture the continuous and auto-correlated nature of eye-gaze and complementary to the approach our computational model uses in integrating eye movements into evidence accumulation. However, as we mentioned in response to Reviewer 1, and now discuss in the paper, our results are robust to other specifications of early eye gaze. For example, if using first fixations (rather than fixation percentages in a specific window) we obtain nearly identical results. We prefer to use a more data-driven approach to identifying relevant time windows, but have now included a mention of this in the paper (see pg. 13) as well as a supplementary note that details these fixation-based analyses (Supplementary Note 1).

In short, the paper seems thorough and important, and I generally support publication, but I found some of the results and modeling opaque. It may be helpful for the authors to provide more detail on how the model supports the overall conclusions of the paper and verify that it is robust out-of-sample. The dropout rate of time-pressure trials is also important to look into.

We once again thank the reviewer for their thoughtful considerations and have incorporated their concerns into our revised manuscript.

References:

- Bogacz, R., Brown, E., Moehlis, J., Holmes, P., & Cohen, J. D. (2006). The physics of optimal decision making: A formal analysis of models of performance in two-alternative forced-choice tasks. *Psychological Review*, *113*(4), 700–765. <https://doi.org/10.1037/0033-295X.113.4.700>
- Bouwmeester, S., Verkoeijen, P. P. J. L., Aczel, B., Barbosa, F., Bègue, L., Brañas-Garza, P., ... Espín, A. M. (2017). Registered replication report: Rand, greene, and nowak (2012). *Perspectives on Psychological Science*, *12*(3), 527–542.
- Chen, F., & Krajbich, I. (2018). Biased sequential sampling underlies the effects of time pressure and delay in social decision making. *Nature Communications*, *9*(1), 3557. <https://doi.org/10.1038/s41467-018-05994-9>
- Diederich, A., & Trueblood, J. S. (2018). A dynamic dual process model of risky decision making. *Psychological Review*, *125*(2), 270.
- Evans, A. M., Dillon, K. D., & Rand, D. G. (2015). Fast But Not Intuitive, Slow But Not Reflective: Decision Conflict Drives Reaction Times in Social Dilemmas. *Journal of Experimental Psychology: General*, *144*(5), 951–966.
- Galizzi, M. M., & Navarro-Martínez, D. (2018). On the external validity of social preference games: a systematic lab-field study. *Management Science*, *65*(3), 976–1002.
- Hawkins, G. E., Forstmann, B. U., Wagenmakers, E.-J., Ratcliff, R., & Brown, S. D. (2015). Revisiting the Evidence for Collapsing Boundaries and Urgency Signals in Perceptual Decision-Making. *Journal of Neuroscience*, *35*(6), 2476–2484. <https://doi.org/10.1523/JNEUROSCI.2410-14.2015>
- Hutcherson, C. A., Bushong, B., & Rangel, A. (2015). A Neurocomputational Model of Altruistic Choice and Its Implications. *Neuron*, *87*(2), 451–463. <https://doi.org/10.1016/j.neuron.2015.06.031>
- Kahneman, D., & Egan, P. (2011). *Thinking, fast and slow*. Farrar, Straus and Giroux New York.
- Krajbich, I. (2018). Accounting for attention in sequential sampling models of decision making. *Current Opinion in Psychology*.
- Krajbich, I., Armel, C., & Rangel, A. (2010). Visual fixations and the computation and comparison of value in simple choice. *Nature Neuroscience*, *13*(10), 1292–1298. <https://doi.org/10.1038/nn.2635>
- Krajbich, I., Bartling, B., Hare, T., & Fehr, E. (2015). Rethinking fast and slow based on a critique of reaction-time reverse inference. *Nature Communications*, *6*, 7455.
- Krawczyk, M., & Sylwestrzak, M. (2018). Exploring the role of deliberation time in non-selfish behavior: The double response method. *Journal of Behavioral and Experimental Economics*, *72*, 121–134. <https://doi.org/https://doi.org/10.1016/j.socec.2017.12.004>
- Lohse, J. (2016). Smart or selfish – When smart guys finish nice. *Journal of Behavioral and Experimental Economics*, *64*, 28–40. <https://doi.org/10.1016/j.socec.2016.04.002>
- Lohse, J., Goeschl, T., & Diederich, J. (2014). Giving is a Question of Time: Response Times and Contributions to a Real World Public Good. *University of Heidelberg, Department of Economics Discussion Paper Series*, (566), 1–20. <https://doi.org/10.2139/ssrn.2457905>
- Milosavljevic, M., Malmaud, J., Huth, A., Koch, C., & Rangel, A. (2010). The Drift Diffusion Model Can Account for the Accuracy and Reaction Time of Value-Based Choices Under High and Low Time Pressure. *Judgment and Decision Making*, *5*(6), 437–449.
- Mischkowski, D., & Glöckner, A. (2016). Spontaneous cooperation for prosocials, but not for

- proselfs: Social value orientation moderates spontaneous cooperation behavior. *Scientific Reports*, 6(1), 21555. <https://doi.org/10.1038/srep21555>
- Moore, D. a., & Loewenstein, G. (2004). Self-interest, automaticity, and the psychology of conflict of interest. *Social Justice Research*, 17(2), 189–202. <https://doi.org/10.1023/B:SORE.0000027409.88372.b4>
- Murphy, R. O., Ackermann, K., & Handgraaf, M. J. J. (2011). Measuring Social Value Orientation. *Judgment and Decision Making*, 6(8), 771–781. <https://doi.org/10.2139/ssrn.1804189>
- Olschewski, S., Rieskamp, J., & Scheibehenne, B. (2018). Taxing cognitive capacities reduces choice consistency rather than preference: A model-based test. *Journal of Experimental Psychology: General*, 147(4), 462–484. <https://doi.org/10.1037/xge0000403>
- Orquin, J. L., & Mueller Loose, S. (2013). Attention and choice: A review on eye movements in decision making. *Acta Psychologica*, 144(1), 190–206. <https://doi.org/10.1016/j.actpsy.2013.06.003>
- Piovesan, M., & Wengström, E. (2009). Fast or fair? A study of response times. *Economics Letters*, 105(2), 193–196.
- Rand, D. G. (2016). Cooperation, Fast and Slow: Meta-Analytic Evidence for a Theory of Social Heuristics and Self-Interested Deliberation. *Psychological Science*, 27(9). <https://doi.org/10.1177/0956797616654455>
- Rand, D. G. (2017). Reflections on the Time-Pressure Cooperation Registered Replication Report. *Perspectives on Psychological Science*, 12(3), 543–547. <https://doi.org/10.1177/1745691617693625>
- Rand, D. G., Greene, J. D., & Nowak, M. A. (2012). Spontaneous giving and calculated greed. *Nature*, 489(7416), 427–430. <https://doi.org/10.1038/nature11467>
- Rand, D. G., Newman, G. E., & Wurzbacher, O. M. (2015). Social Context and the Dynamics of Cooperative Choice. *Journal of Behavioral Decision Making*, 28(2), 159–166. <https://doi.org/10.1002/bdm.1837>
- Rand, D. G., Peysakhovich, A., Kraft-Todd, G. T., Newman, G. E., Wurzbacher, O., Nowak, M. A., & Greene, J. D. (2014). Social heuristics shape intuitive cooperation. *Nature Communications*, 5, 1–12. <https://doi.org/10.1038/ncomms4677>
- Shalvi, S., Eldar, O., & Bereby-Meyer, Y. (2012). Honesty Requires Time (and Lack of Justifications). *Psychological Science*, 23(10), 1264–1270. <https://doi.org/10.1177/0956797612443835>
- Smith, S. M., & Krajbich, I. (2019). Gaze Amplifies Value in Decision Making. *Psychological Science*. <https://doi.org/10.1177/0956797618810521>
- Tajima, S., Drugowitsch, J., & Pouget, A. (2016). Optimal policy for value-based decision-making. *Nature Communications*, 7, 1–12. <https://doi.org/10.1038/ncomms12400>
- Van Lange, P. A. M. (1999). The pursuit of joint outcomes and equality in outcomes: An integrative model of social value orientation. *Journal of Personality and Social Psychology*, 77(2), 337.
- Yamagishi, T., Matsumoto, Y., Kiyonari, T., Takagishi, H., Li, Y., Kanai, R., & Sakagami, M. (2017). Response time in economic games reflects different types of decision conflict for prosocial and proself individuals. *Proceedings of the National Academy of Sciences*, 114(24), 6394–6399. <https://doi.org/10.1073/pnas.1608877114>
- Zaki, J., & Mitchell, J. P. (2013). Intuitive Prosociality. *Current Directions in Psychological Science*, 22(6), 466–470. <https://doi.org/10.1177/0963721413492764>

Reviewers' comments:

Reviewer #1 (Remarks to the Author):

The authors have done an excellent job in the revision and the paper has substantially improved. By adding further analyses and model comparisons, further discussions and an additional study, they addressed all but one my concerns. Also, it is great that the new study used state of the art pre-registration, although other techniques would have been available to better manipulate attention (eye-tracking studies like used in most of the recent work: Ghaffari & Fiedler, 2018, PsySci).

In some points, I am not fully following the author's arguments, but these points concern issues that should probably be discussed not in a review process but in an open debate in future publications.

There are a few remaining issues that should be addressed:

- The authors misunderstood my second point: "2) Internal validity: One alternative explanation for the behavioral results would be that the 3 spontaneous responses are more noisy and people who behave more extremely show regression to the mean if more time is available." – I do not doubt that results are more noisy under time pressure (as the results in the authors responses suggest) – but my argument is that if the authors select /classify persons based on extreme responses in a noisy environment, there will be a stronger regression to the mean effect if they measure the same variables again. The authors have to address this major point in the main text and to rule it out (perhaps they can with the new study). My suggestion to use an independent measure of social preferences (e.g. SVO measure) pointed in the same direction: when the person's classification is based on an independent measure, the point can be ruled out.

- The authors note that they will provide the code and the data (only on) reasonable request. According to current standards of transparency, they should be made directly available at OSF.

- For the new study, the frequency with which all information was inspected should be reported in the main text and the analysis should also be run with only the trials in which all pieces of information are inspected. It is crucial that the results are not only driven by the fact that people do not look at the other outcomes.

- The findings of the Ghaffari & Fiedler (2018) paper should be discussed since it provides the most similar work to the newly included study that manipulates attention to influence social preferences.

Ghaffari, M., & Fiedler, S. (2018). The Power of Attention: Using Eye Gaze to Predict Other-Regarding and Moral Choices. *Psychological Science*, 29(11), 1878-1889.

Reviewer #2 (Remarks to the Author):

The authors have fully addressed my comments on the manuscript. The detailed and thoughtful replies are appreciated!

Reviewer #3 (Remarks to the Author):

I thank the authors for their very comprehensive replies to my comments. It was particularly thoughtful of them to run a follow-up study to address one of my concerns. I have no lingering worries about the paper and believe it would make an excellent contribution to the journal.

Reviewers' comments:

Reviewer #1 (Remarks to the Author):

The authors have done an excellent job in the revision and the paper has substantially improved. By adding further analyses and model comparisons, further discussions and an additional study, they addressed all but one my concerns. Also, it is great that the new study used state of the art pre-registration, although other techniques would have been available to better manipulate attention (eye-tracking studies like used in most of the recent work: Ghaffari & Fiedler, 2018, PsySci).

In some points, I am not fully following the author's arguments, but these points concern issues that should probably be discussed not in a review process but in an open debate in future publications.

We thank the reviewer for their encouragement and will incorporate their suggestions for references to relevant literature. We very much look forward to open debate suggested, and hope it will lead to a better understanding of these interesting phenomena!

There are a few remaining issues that should be addressed:

- The authors misunderstood my second point: "2) Internal validity: One alternative explanation for the behavioral results would be that the 3 spontaneous responses are more noisy and people who behave more extremely show regression to the mean if more time is available." – I do not doubt that results are more noisy under time pressure (as the results in the authors responses suggest) – but my argument is that if the authors select /classify persons based on extreme responses in a noisy environment, there will be a stronger regression to the mean effect if they measure the same variables again. The authors have to address this major point in the main text and to rule it out (perhaps they can with the new study). My suggestion to use an independent measure of social preferences (e.g. SVO measure) pointed in the same direction: when the person's classification is based on an independent measure, the point can be ruled out.

We apologize for our misunderstanding of the reviewer's original comment. We now fully appreciate and agree with the reviewer's concern. However, we think it is unlikely that the effects here are driven by regression to the mean. One of the key issues determining the likelihood of observing a significant regression to the mean effect is the extent of measurement variability: the higher the variability, the more likely regression to the mean. In this regard, while we describe generosity as a single measure in the two separate conditions, we wish to note that it is actually a composite across many repeated measurements (80 trials per condition). Hence, the mean estimates of generosity in either condition are likely to be relatively close to the "true estimates" (Barnett, Van Der Pols, & Dobson, 2005) given the small possibility that random within-individual variability could result in consistent selfish/generous responding across 80 trials. To illustrate this point, we find excellent reliability across blocks (20 trials each) in generosity estimates for each condition:

Table: Reliability of generosity estimates across blocks for each condition in 3 studies.

Study	Number of blocks	Cronbach's α [95% CI]	
		High time pressure	Low time pressure
Study 1	4	0.94 [0.91, 0.96]	0.92 [0.90, 0.96]
Replication Study 1	5	0.93 [0.91, 0.96]	0.92 [0.90, 0.95]
Replication Study 2	5	0.95 [0.93, 0.97]	0.95 [0.92, 0.97]

To get some sense of the likelihood that our effects are simply due to regression to the mean, we re-ran the same correlational analyses we performed on the full set of trials investigating the association between generosity under high time pressure and change in generosity with time, but now separately for each block. We find overwhelming consistency in the pattern of effects across all blocks (Study 1: mean Pearson's $r = -0.398$, all $ps < .01$; Replication Study 1: mean Pearson's $r = -0.455$, all $ps < .05$; Replication Study 2: mean Pearson's $r = -0.390$, all $ps < .05$) Thus, we think it is highly unlikely that random variability in estimates resulting in regression to the mean would exhibit such consistency across multiple tests and experiments. Together, these new analyses strongly suggest that individuals do exhibit a unique pattern of choice biases under time pressure that are mitigated with time.

We also performed an even stronger test of regression to the mean vs. our favoured hypothesis. Assumptions of regression to the mean suggest that the strongest drivers of the effect would be the most extreme predictors (since these are the observations that can shift the most towards the mean). In contrast, our prioritized attention model suggests that the most extreme individuals (ones who attend 100% of the time to self or to other exclusively) should actually be *least* likely to shift in a more moderate direction, since they are better able to implement their true preferences. It should actually be the individuals who give at least *some* weight to the secondary attribute, and attend to it when given enough time, who show the most change. Thus, the regression to the mean explanation suggests that observed associations between generosity and change due to time pressure should get weaker when the most extreme individuals are excluded, whereas our model predicts that this association should stay the same, or if anything get stronger. To test these two distinct possibilities, we ran a follow-up analysis excluding participants who were $< 25\%$ generous or $> 75\%$ generous under time pressure. Instead of attenuating the original effect (Study 1: Pearson's $r = -0.313$, $t_{58} = 2.513$, $p = .0148$; Replication Study 1: Pearson's $r = -0.286$, $t_{63} = -2.3702$, $p = .021$; Replication Study 2: Pearson's $r = -0.300$, $t_{47} = -2.1537$, $p = .036$), we find that the effect becomes stronger when excluding the participants most likely to regress towards the mean (Study 1: Pearson's $r = -0.490$, $t_{35} = -3.326$, $p = .0021$; Replication Study 1: Pearson's $r = -0.408$, $t_{52} = -3.221$, $p = .0022$; Replication Study 2: Pearson's $r = -0.522$, $t_{32} = -3.459$, $p = .0016$).

This effect is fully consistent with our subsequent analyses that show individuals' gaze biases to be particularly important in mediating this effect. For example, extremely selfish individuals typically make consistently selfish choices under low time pressure as an expression of their selfish social preferences. Under high time pressure, they are also likely to search exclusively for their own outcomes first, ensuring that their selfish preferences are sustained and expressed. Thus, we observe no change in their behavior. In contrast, moderately selfish individuals, while still preferring to maximize their own outcomes, also give some attention to others' outcomes under low time pressure. Under high time pressure however, because they are more likely to search for their own outcomes first, and then have little time to acquire and process the other

person's outcomes, their choices become more extreme reflections of their underlying preferences. Given these arguments, we do not believe it is necessary to employ an independent measure of prosociality, like the SVO, to illustrate this point.

Finally, our model cross-validation strongly suggests that the individual-level parameters estimated from half of the data is sufficient to predict patterns of change in choice behavior under time pressure in an out-of-sample dataset. These systematic associations between social preferences, attention and choice would be highly unlikely in an account of choice that simply assumes regression to the mean. We now report these analyses as part of supplementary note 2 and make mention of it in the main paper in the results section (Pg 7).

- The authors note that they will provide the code and the data (only on) reasonable request. According to current standards of transparency, they should be made directly available at OSF.

While we agree with the reviewer and editor that direct access to all code and data would be ideal for purposes of transparency, the data reported in these studies were collected at a time when such practice was not yet standard (which we note with amazement was not such a long time ago!). As a result, the ethics protocols under which this data was collected only permits sharing of participants' data upon reasonable request to the author. We will submit an amendment to the relevant ethics committee to obtain permission to make the de-identified data available on an OSF repository as requested. Should this amendment be approved before publication of the paper, we will update all relevant references to include a link to the data. The code will be made publicly available at the following link: <https://osf.io/vf6a5/> following acceptance of the manuscript and included this link on pg 33 of the main paper. Here is a corresponding view-only link to the repository: https://osf.io/vf6a5/?view_only=d0ca47b6abe441b59765f29df9611272.

- For the new study, the frequency with which all information was inspected should be reported in the main text and the analysis should also be run with only the trials in which all pieces of information are inspected. It is crucial that the results are not only driven by the fact that people do not look at the other outcomes.

We agree with the reviewer that it is important to know which of the multiple possible mechanisms through which attention interacts with time pressure to affect the decision-making process. It is in fact *precisely* this point that we are trying to make: that time pressure can both amplify attentional biases to acquire certain pieces of information search, and then can also truncate the search process before all information has been fully comprehended. As we state on pg. 4, "Such a model is consistent with research showing that selfish individuals attend more to information about self-interest while prosocial individuals attend more to information about the welfare of others, but goes a step further in suggesting that time pressure (which may force individuals to make fast rather than fully-informed choices) should amplify the strategic deployment of attention towards information prioritized by the individual." Thus, our model actually explicitly predicts that the primary driver of behaviour change under time pressure should be the combination of where an individual looks first (due either to endogenous preferences or exogenous cues) and whether they acquire any further information. Our model

further suggests that people should be more strategic under time pressure, both about what information they acquire first, as well as whether and when they acquire additional information. The results of both Studies 1 and 2 are fully in line with this theory.

In our second study, we find that participants were indeed less likely to inspect all information under high time pressure ($M_{\text{high}} = 75.32\%$) than low time pressure ($M_{\text{low}} = 93.02\%$). Mixed effects logistic regressions revealed this difference to be significant ($b = 2.322$, $SE = 0.0519$, $z = 44.77$, $p < .001$). This effect is also true of Study 1 ($M_{\text{high}} = 87.92\%$; $M_{\text{low}} = 98.00\%$; $b = 2.567$, $SE = 0.1399$, $z = 18.35$, $p < .001$).

Moreover, we find that people show evidence of strategically deciding whether to acquire additional information after their first fixation, especially under time pressure. For example, our model suggests that, if people are selfish, they should feel less need to acquire additional information under time pressure when what they are forced to look at corresponds to their already-preferred information (i.e. self-outcomes), but should be more likely to prioritize additional information search if we force them to look at less-preferred information. This predicts that people should show fewer fixations to distinct information when they acquire their preferred information first, and this should be especially true under time pressure, which makes information acquisition more costly. Mixed effects generalized-poisson regression on number of fixations as a function of time pressure and first fixation confirmed this prediction. Participants were more likely to make fewer fixations in a trial if they looked at self-outcomes first (Low time pressure: $b = -0.034$, $SE = 0.005$, $z = -6.88$, $p < .001$), and this effect was significantly stronger under time pressure (High time pressure: $b = -0.131$, $SE = 0.006$, $z = -21.81$, $p < .001$; Interaction: $b = -0.097$, $SE = 0.008$, $z = -12.63$, $p < .001$).

Figure: First fixation position and time pressure interact to predict the total number of fixations on a trial. The middle line of the box-plot indicates the group mean number of fixations per trial with the upper and lower boundaries of the box indicating $\pm 1SE$. The violin plots indicate the distribution of participants' mean number of fixations per trial.

Together, these analyses suggest that individuals were more likely to expedite their choices at the expense of incomplete information search under high time pressure. Furthermore, participants sacrificed additional information in a strategic way such that they were more likely to make choices based on incomplete information when they had already acquired what we assume to generally be the highest-priority information, namely, their own outcomes.

This findings are fully in line with one of the main aims of our paper, which is to show that the effects of time pressure on generous choice may be mediated through strategic shifts in attention, and that these strategic aspects of choice may provide a better, more parsimonious alternative explanation that help to resolve the existing conflicts in the broader literature. Given this scope, we think that the observation that time pressure induces biased and incomplete information search is actually supportive of our arguments regarding the dynamic role of attention and its influence during choice behavior under constraints.

However, we also share an interest with the reviewer in knowing whether our results are *fully* explained by what we might think of as time-pressure induced ignorance, or also result from incomplete incorporation of that information due to attention. This question relates to existing reviews of the literature on attention and choice, which have suggested that the main *causal influence* of attention on choice is its role in the initial acquisition of relevant information (see: Orquin & Mueller Loose, 2013).

To address this question, we ran a third on-line experiment in which we presented information serially in the dictator game and interrupted participants at different points in their viewing to elicit their choice. In each trial, participants (N = 103) were first presented either their own outcomes (\$Self) or their partner's outcomes (\$Other) for ~300ms. They were then presented the other piece of information (\$Other or \$Self). In the first three conditions, participants were either presented the second piece of information for 100ms, 200ms or 400ms before they were interrupted and prompted to make their choice. In the other two-conditions (repeated, and repeated-long), participants were presented both pieces of information twice. In the repeated condition, participants were presented with information in the following order: 1st (~300ms), 2nd (~300ms), 1st (~300ms), 2nd (200ms) before being prompted to make a response. In the repeated-long exposure condition, participants were also presented with both pieces of information twice but for longer durations, such that they are presented with information in the following order: 1st (~500ms), 2nd (~500ms), 1st (~500ms), 2nd (500ms) before prompted to make a response.

Since, in all conditions, participants were exposed to both information: \$Self and \$Other, the ignorance model of attention would predict no effect of attention on choice while the attentional amplification model would predict that the focus of early attention would bias choice but its effects should weaken as people sample more information. Additional preregistered experimental details, hypotheses and analyses can be found on OSF at the following link: https://osf.io/hw9um/?view_only=6c70927f1f144b20b96f18cac1808958.

Importantly, using mixed-effects logistic regressions predicting binary generous choice, we find an interaction between the first information presented (first fixation) and 2nd information exposure conditions (Wald's $\chi^2(4) = 20.621$, $p < .001$), such that first fixations predicted

generosity when exposure to the 2nd piece of information was relatively short (100ms: $b = -0.476$, $SE = 0.0857$, $z = -5.560$, $p < .001$; 200ms: $b = -0.209$, $SE = 0.0863$, $z = -2.416$, $p = .0157$) but not when participants were exposed to the 2nd information for extended periods of time (400ms: $b = -0.074$, $SE = 0.0898$, $z = -0.828$, $p = .407$; repeated: $b = -0.086$, $SE = 0.0889$, $z = -0.972$, $p = .331$; repeated-long: $b = 0.040$, $SE = 0.0887$, $z = 0.457$, $p = .648$). This pattern of results not only replicate findings in Study 2, but also suggest that ignorance cannot fully explain the effects of early attentional biases on choice.

Figure: First fixation position and 2nd information exposure condition interact to predict the total number of fixations on a trial. The middle line of the box-plot indicates the group mean number of fixations per trial with the upper and lower boundaries of the box indicating +/- 1SE and the whiskers indicating the 95% confidence interval.

Although we think these results are interesting, and support *both* the effects of ignorance and the effects of attention on time-pressure induced biases, we believe they deserve their own treatment and have highlighted this in the discussion of our paper. Given that the paper is already somewhat long as is, we have therefore currently opted not to include an extensive discussion of them in the current article. However, if the editor or review feels that this would be crucial to include, we can of course do so, either in the main body of the paper or in a supplementary note.

Results: The Gaze-Informed Attentional Drift Diffusion Model

“Unlike previous applications of the ADDM, which model attention generically using group or trial averages (Krajbich, Armel, & Rangel, 2010; Krajbich & Rangel, 2011; Smith & Krajbich, 2019), our model makes full use of each individual’s moment-by-moment gaze position to determine exactly when different attributes (1) enter consideration (Orquin & Mueller Loose, 2013) and (2) receive amplification.”

Discussion:

“Furthermore, the exact mechanisms of attention’s influence on the evidence-accumulation process remains unclear. Some work suggests that attention supports the initial construction of the choice-set while others posit that attention amplifies the value of evidence in real-time. While we have included both these possibilities as attentional mechanisms in our computational model, the field continues to debate the relative contributions of these mechanisms during choice (Ghaffari & Fiedler, 2018; Orquin & Mueller Loose, 2013; Smith & Krajbich, 2019). Future work should seek to characterize and disambiguate between these mechanisms and their downstream effects on choice. We suspect that neurocomputational models like the one we have developed here will be key for resolving such issues.”

- The findings of the Ghaffari & Fiedler (2018) paper should be discussed since it provides the most similar work to the newly included study that manipulates attention to influence social preferences.

Ghaffari, M., & Fiedler, S. (2018). *The Power of Attention: Using Eye Gaze to Predict Other-Regarding and Moral Choices*. *Psychological Science*, 29(11), 1878-1889.

We thank the reviewer for the reference. We have since included a reference to this paper in the overall discussion (Pg 21).

References:

- Barnett, A. G., Van Der Pols, J. C., & Dobson, A. J. (2005). Regression to the mean: what it is and how to deal with it. *International Journal of Epidemiology*, 34(1), 215–220.
- Ghaffari, M., & Fiedler, S. (2018). The power of attention: Using eye gaze to predict other-regarding and moral choices. *Psychological Science*, 29(11), 1878–1889.
- Krajbich, I., Armel, C., & Rangel, A. (2010). Visual fixations and the computation and comparison of value in simple choice. *Nature Neuroscience*, 13(10), 1292–1298.
<https://doi.org/10.1038/nn.2635>
- Krajbich, I., & Rangel, A. (2011). Multialternative drift-diffusion model predicts the relationship between visual fixations and choice in value-based decisions. *Proceedings of the National Academy of Sciences*, 108(33), 13852–13857. <https://doi.org/10.1073/pnas.1101328108>
- Orquin, J. L., & Mueller Loose, S. (2013). Attention and choice: A review on eye movements in decision making. *Acta Psychologica*, 144(1), 190–206.
<https://doi.org/10.1016/j.actpsy.2013.06.003>
- Smith, S. M., & Krajbich, I. (2019). Gaze Amplifies Value in Decision Making. *Psychological Science*. <https://doi.org/10.1177/0956797618810521>

Reviewer #2 (Remarks to the Author):

The authors have fully addressed my comments on the manuscript. The detailed and thoughtful replies are appreciated!

We thank the reviewer for their encouraging remarks.

Reviewer #3 (Remarks to the Author):

I thank the authors for their very comprehensive replies to my comments. It was particularly thoughtful of them to run a follow-up study to address one of my concerns. I have no lingering worries about the paper and believe it would make an excellent contribution to the journal.

We thank the reviewer for their encouraging remarks.

****REVIEWERS' COMMENTS:**

Reviewer #1 (Remarks to the Author):

I thank the authors for comprehensively addressing my remaining points in the second revision. The further analyses clearly rule out my concerns about regression to the mean effects. The further analyses concerning ignorance vs. attention mechanisms are enlightening and convincing too.

This is a great paper and I recommend publication.